# Modular (de)construction of complex bacterial phenotypes by CRISPR/nCas9-assisted, multiplex cytidine base-editing

Daniel C. Volke[1,4], Román A. Martino[2,3,4], Ekaterina Kozaeva[1], Andrea M. Smania[2,3] & Pablo I. Nikel [1✉]

CRISPR/Cas technologies constitute a powerful tool for genome engineering, yet their use in non-traditional bacteria depends on host factors or exogenous recombinases, which limits both efficiency and throughput. Here we mitigate these practical constraints by developing a widely-applicable genome engineering toolset for Gram-negative bacteria. The challenge is addressed by tailoring a CRISPR base editor that enables single-nucleotide resolution manipulations (C·G → T·A) with >90% efficiency. Furthermore, incorporating Cas6-mediated processing of guide RNAs in a streamlined protocol for plasmid assembly supports multiplex base editing with >85% efficiency. The toolset is adopted to construct and deconstruct complex phenotypes in the soil bacterium *Pseudomonas putida*. Single-step engineering of an aromatic-compound production phenotype and multi-step deconstruction of the intricate redox metabolism illustrate the versatility of multiplex base editing afforded by our toolbox. Hence, this approach overcomes typical limitations of previous technologies and empowers engineering programs in Gram-negative bacteria that were out of reach thus far.

[1] The Novo Nordisk Foundation Center for Biosustainability, Technical University of Denmark, Kongens Lyngby, Denmark. [2] Departamento de Química Biológica Ranwel Caputto, Facultad de Ciencias Químicas, Universidad Nacional de Córdoba, Córdoba, Argentina. [3] Centro de Investigaciones en Química Biológica de Córdoba (CIQUIBIC), CONICET, Universidad Nacional de Córdoba, Córdoba, Argentina. [4]These authors contributed equally: Daniel C. Volke, Román A. Martino. ✉email: pabnik@biosustain.dtu.dk

Technologies based on CRISPR (clustered regularly inter-spaced short palindromic repeats)/Cas (CRISPR-associated proteins) have been repurposed for genome modification, mediating a true revolution in synthetic biology and metabolic engineering[1–4]. With numerous applications in diverse prokaryotic and eukaryotic species, CRISPR/Cas approaches continue to facilitate the design and construction of synthetic organisms[5,6]. DNA base editors (BEs), comprising a deaminase fused to a catalytic inactive or partly active Cas protein, are DNA editing tools specifically adopted for establishing genome-wide specific modifications[7–9]. While the mutant Cas protein retains its ability to bind and unwind double-stranded DNA, it cannot cleave both DNA strands. Some BEs include a *nicking Cas9*, which only cleaves ('nicks') one of the DNA strands; others rely on a Cas protein that has lost the ability of cleaving DNA altogether. Regardless of this feature, the BE is guided to the specific target locus by a short RNA motif, a *guide RNA* (gRNA). A 20-nucleotide (nt)-long stretch in the gRNA, the spacer, forms a heteroduplex with the target DNA strand, the protospacer. Thereby, the complex exposes the second DNA strand, rendering it available for the deaminase. BEs are classified as cytidine base editors (CBE) or adenine base editors (ABE). The deaminase in a CBE converts a cytosine in the exposed strand into uracil, which is subsequently repaired to thymine by native enzymes during DNA replication[10]. In an ABE, in contrast, the deaminase acts on adenine to yield guanine[11]. BEs have been shown to operate in a broad range of organisms[11–16], including mammalian cells, yeast, and bacterial species. However, the scope and utility of BEs offer ample room for optimization—e.g., for metabolic engineering applications with non-traditional bacterial hosts, such as *Pseudomonas*.

*Pseudomonas* species continue to attract attention both in fundamental and applied research. Some clinically-relevant species, e.g., the opportunistic pathogen *Pseudomonas aeruginosa*, constitute a frequent cause of nosocomial complications in immunosuppressed patients[17,18]. Other *Pseudomonas* are plant pathogens that can infect a variety of commercial crops[19]. Moreover, *Pseudomonas* tolerates high levels of abiotic stress, including oxidative damage and solvents, underscoring their potential as biotechnological hosts for large-scale fermentations[20]. *Pseudomonas putida* is endowed with a rich and versatile metabolism[21] that renders this species an ideal platform for metabolic engineering towards the production of chemicals[22–25]. The biochemical repertoire of *P. putida* comprises enzymes for the assimilation of complex carbon sources, e.g., lignocellulosic hydrolysates, and for the biosynthesis of a broad range of chemical building-blocks and polymers[26,27]. Genome engineering tools have been key to advance *Pseudomonas* research, linking phenotypes to genotypes and enabling the investigation of the metabolic impact of genome alterations. Tools for gene deletion, insertion, and genome modification are established for *P. putida*[28–32], yet the techniques currently used are tedious, time-consuming, and difficult-to-automate[33,34]. Importantly, the simultaneous introduction of multiple modifications at distal chromosome loci has been implemented in just a few biotechnology workhorses, e.g., *Escherichia coli*[35,36] and yeast[37]. For instance, λ-Red recombineering, commonly used for *E. coli* engineering[38], does not operate efficiently in *Pseudomonas*[39]—likely due to limited activity of phage recombinases in these species, or because native proteins interfere with their action. While CRISPR/Cas toolsets accelerated microbial engineering efforts[40,41], non-homologous end joining (NHEJ) mechanisms are virtually absent in prokaryotes[42], preventing the introduction of frameshifts in target loci—and functional knock-outs thereof. A recent proof-of-principle study on base editing in *Pseudomonas* by Chen et al[41]. reported promising results. Considering this

background, could BEs fill the gap towards fast, reliable, and multiplexed genome engineering of *Pseudomonas*?

Here, we developed a multiplexed CBE technology for Gram-negative bacterial species that enables the fastest modification of *Pseudomonas* genomes reported thus far. Starting by a thorough characterization of the exact editing window, together with the influence of incubation times, the editing performance of a synthetic CBE was further enhanced by engineering a uracil glycosylase inhibitor (*ugi*) in the system. Additionally, we investigated the positional effect of the gRNAs in the RNA cassette and the processing effect of a synthetic Cas6 element on the editing efficiency. Furthermore, we demonstrate multiplex genome editing with a streamlined plasmid assembly protocol that renders the reliable cloning of plasmids with >10 genome targets possible. This CBE toolbox enabled the one-step engineering of a *P. putida* strain tailored for high-production of the added-value platform chemical protocatechuic acid (PCA, 3,4-dihydroxybenzoic acid) from sugars. Additionally, a stepwise editing approach was adopted to disentangle the complex redox metabolism in *P. putida*, which led to a NADPH-depleted strain that can be used for screening NADPH-producing activities—both native and heterologous. This CBE toolbox enables the construction and deconstruction of complex phenotypes in both model hosts and non-traditional bacteria—including genome engineering programs that have been virtually impossible to implement using the currently-available genome engineering technologies.

## Results

**Base-editing as a feasible tool to introduce genome-wide functional knock-outs in *Pseudomonas*.** A CBE converts cytidine into thymine (C·G → T·A) in target regions of the genome (Fig. 1A). This alteration, ultimately fixed in the genome during DNA repair and replication (Fig. 1B), can be harnessed to introduce premature *STOP* codons in open reading frames (ORFs) that lead to truncated, non-functional proteins—thus mediating functional gene knock-outs. Several requisites have to be met by the targeted locus to become a suitable candidate for base editing. Firstly, the protospacer adjacent motif (PAM) has to be located within a relatively narrow window close to the targeted cytidine. The optimal distance between the PAM and the cytidine residue[8] seems to be 13–19 bp. The PAM sequence for (n)Cas9 from *Streptococcus pyogenes* is 5′-NGG-3′ (*N* represents any nucleotide), a motif frequently present in the GC-rich chromosome of *Pseudomonas* species[21]. Secondly, the cytidine-to-thymidine change should lead to a *STOP* codon as close as possible to the *START* codon of the coding sequence. Thirdly, no guanidine residue should be located immediately upstream of the targeted cytidine, as this base sterically hinders the editor and could reduce editing efficiency. Finally, the protospacer sequence has to be unique in the genome to circumvent off-target editing events.

Based on these constraints, we scanned all ORFs of two model species of the broad *Pseudomonas* genus (i.e., *P. putida* KT2440 and *P. aeruginosa* PA14, representative strains biotechnologically- and clinically-relevant bacteria) to identify potential targets for CBE-mediated modification. An algorithm was developed to automate the scanning procedure by uploading any bacterial genome of interest and running a script that locates target cytidine residues subjected to the conditions listed above (Supplementary Data 1). According to this analysis, 5059 of all ORFs in *P. putida* KT2440 (92%) could be targeted by introducing a premature *STOP* codon (Fig. 1C), with ca. half of all ORFs accessible for modification within the first 20% of the coding sequence. We found that >2000 genes could be interrupted by introducing a *STOP* codon within the first 10%

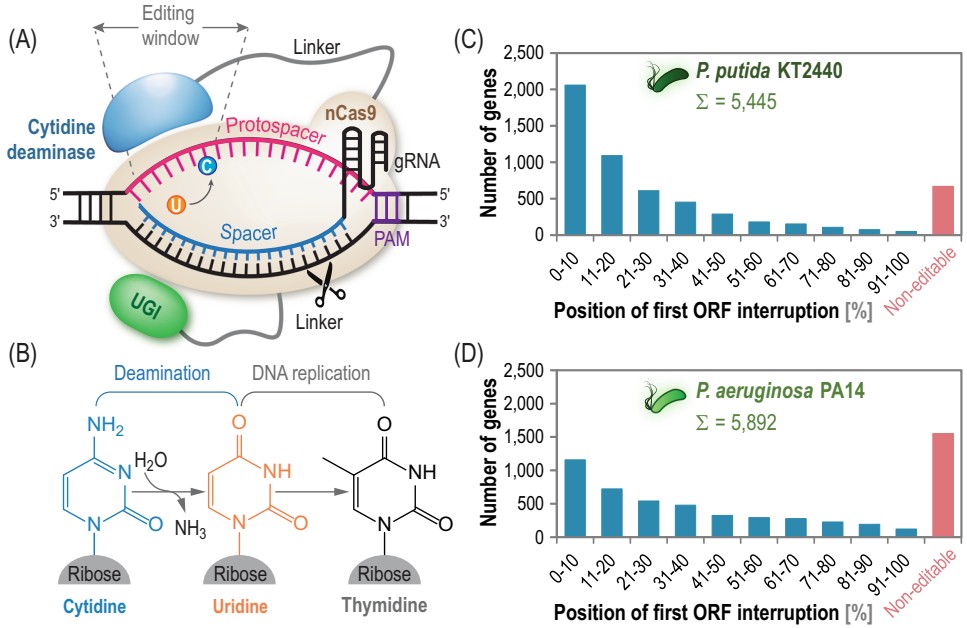

**Fig. 1 Mechanism of cytidine modification and editing scope in *Pseudomonas* species. A** Structure of the cytidine base editor (CBE). A nicking Cas9 (nCas9) is guided by a single guide RNA (sgRNA) to the cognate protospacer sequence. Upon forming a heteroduplex between the spacer region of the gRNA and its target sequence, the complementary DNA strand becomes exposed. A cytidine deaminase, fused to nCas9, gains access to the exposed strand and converts cytosine to uracil within an editing window; uridine is subsequently transformed into thymidine during DNA replication. The uracil DNA glycosylase inhibitor (UGI) prevents repair of the deaminated cytidine by the endogenous DNA repair machinery. *PAM*, protospacer adjacent motif. **B** Deamination of a cytidine residue subsequently leads to uridine and thymidine. **C** In silico genome analysis of open reading frames (*ORFs*) annotated in *P. putida* KT2440 and **D** *P. aeruginosa* PA14 reveals the scope of base editing across the bacterial chromosomes. The CBE can be used to interrupt ca. 92 and 75% of all ORFs in *P. putida* and *P. aeruginosa* PA14, respectively, by introducing *STOP* codons at the positions indicated. Source data underlying panels **C**, **D** are provided as a Source Data file.

of the ORF after the *START* codon. In contrast, <8% of the ORFs annotated in the genome of strain KT2440 were not amenable to CBE-mediated modification. The analysis was repeated for *P. aeruginosa* PA14 (Fig. 1D). In this case, 75% of all ORFs are accessible for base editing, and only 1,553 genes cannot be modified using CBEs. This in silico analysis can be extended to any bacterial genome (*Pseudomonas* or otherwise) available in public databases[43], and our script also allows for scanning of alternative, non-canonical PAMs[44] (e.g., 5′-NGA-3′). We concluded that the CBE constitutes a powerful tool for generating functional knock-outs for most genes in *Pseudomonas* species, and we set out to construct a standardized CBE toolbox to this end.

**Design of a set of standard plasmids supporting highly-efficient multiplex genome editing in Gram-negative bacterial species.** A full characterization and debottlenecking of the base-editing tool started by investigating the effect of UGI on the editing efficiency. UGI is a small (9.5 kDa) protein isolated from *Bacillus subtilis* bacteriophage PBS1 that inhibits uracil-DNA glycosylases (UNGs; Fig. 2A). Owing to their critical role in DNA repair, UNGs are ubiquitous enzymes across all domains of Life. UNG catalyzes the excision of deoxyuracil, the base-editing intermediate[45]—thus reducing the editing efficiency. Introducing UGI in CBEs increased editing efficiency in mammalian cells and *E. coli*[46], but could potentially raise the occurrence of off-target events, i.e., mutations outside the protospacer region. We constructed a set of modular plasmids that contain both (i) the gene encoding the APOBEC1 (apolipoprotein B mRNA editing enzyme, catalytic polypeptide-like) cytidine deaminase from *Rattus norvegicus* (*cda*) as an *N*-terminal fusion with the *S.*

*pyogenes* Cas9 nickase (nCas9, *Sp*Cas9[D10A]) protein via a flexible, 16 amino acid-long XTEN linker[8], and (ii) the corresponding gRNA expression cassette(s) in a single vector backbone (Fig. 2B). The gRNAs in the constructs described in this study are designed as a chimera of a CRISPR RNA and the *trans*-activating CRISPR RNA, also called single gRNAs (sgRNAs)[47]. The starting point for these constructs was vector pnCasPA-BEC, where the two functional elements described above are constitutively expressed through the activity of the P$_{rpsL}$ (from *P. aeruginosa* PAO1) and P$_{trc}$ (synthetic) promoters, respectively. Other relevant features in this plasmid set are (i) the counter-selectable SacB marker (*sacB*, encoding a levansucrase from *B. subtilis*[48]), which confers sucrose-dependent lethality to facilitate plasmid curing after base-editing procedures, (ii) the broad-host-range *pRO1600* origin of vegetative replication[49–51], and (iii) a set of antibiotic resistance cassettes that can be easily swapped according to the rules of the Standard European Vector Architecture[52].

Next, the *ugi* gene from the *Bacillus* bacteriophage AR9 (codon-optimized for *Pseudomonas*) was engineered in plasmid pBEC*x* to yield an APOBEC1-nCas9-UGI chimeric protein (note that, according to the standard plasmid nomenclature, *x* is a placeholder to indicate the antibiotic resistance cassette). Linking the DNA fragments encoding these functionalities by *USER*-cloning gave rise to a *C*-terminal fusion of UGI to the nCas9 protein, leaving a 4 amino acid linker (SGGS) between both functional modules[8,14]. The pBEC*x* plasmid set was further upgraded to enable multiplex editing of targets in the bacterial chromosome. To this end, the CRISPR-associated endoribonuclease gene *cas6f* (also known as *csy4*) from *P. aeruginosa* PA14 was included in our design. In its natural context, Cas6 generates RNAs in type I-F CRISPR systems (i.e., the *Yersinia pestis* subtype) by cleaving pre-crRNAs at the bottom of stable

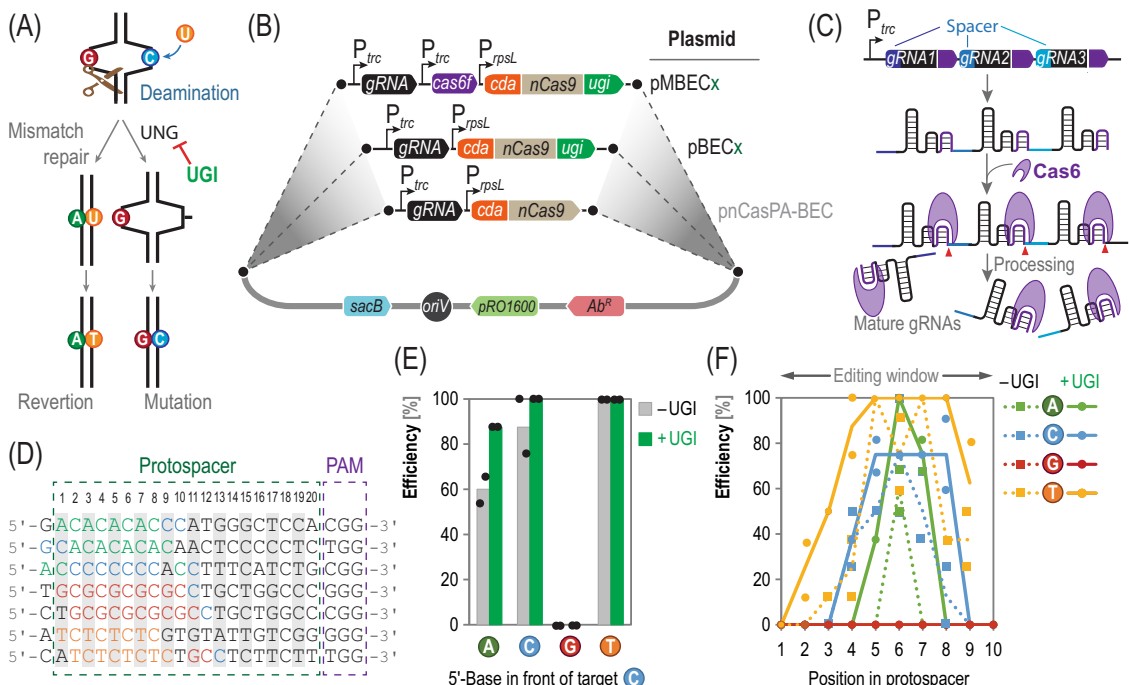

**Fig. 2 Tailoring a cytidine base editor for multiplex genome modifications in *P. putida*. A** Upon cytidine deamination, the paired guanidine is converted to adenine and thymidine, or uridine is recognized by uracil-DNA glycosylase (*UNG*), an activity blocked by the UNG inhibitor (*UGI*). **B** Incorporation of *ugi* to editing plasmids (pBEC) towards increasing editing efficiency. In a subsequent step, *cas6f* was introduced for processing multiple gRNAs (pMBEC). Note that **x** is a placeholder identifying antibiotic resistances (*Ab*R). The vectors are carrying furthermore a vegetative origin of replication (*oriV*) and the sucrose counter selectable marker *sacB*. **C** Processing of the RNA cassette by Cas6. Each gRNA harbors a 3′-recognition site for Cas6; Cas6 remains attached to gRNAs upon cleavage. **D** Protospacers used to investigate positional and preceding base effects. All positions within the editing window are tested for editing with any of the four bases proceeding. *PAM*, protospacer adjacent motif. **E** Editing efficiency at any protospacer position depending on preceding base and **F** editing efficiency depending on position in the protospacer and preceding base, quantified with and without *ugi*. Mean values of two independent biological experiments are presented; dots represent data per experiment with at least eight colonies analyzed per replicate. Source data underlying panels **E**, **F** are provided as a Source Data file.

stem-loops encoded by CRISPR repeats[53]. Cas6 recognizes a short RNA loop and cuts the molecule directly downstream and, as shown by Tsai et al.[54], this endoribonuclease can be harnessed to process a single RNA molecule into several individual gRNAs (Fig. 2C). This strategy offers the advantage that a single promoter is used to drive the expression of several gRNAs, simplifying the design and construction of editing plasmids. Accordingly, *cas6f* (ORF *PA14_33300*) was placed under transcriptional control of the synthetic P*trc* promoter in the pBEC*x* backbone by *USER*-cloning to yield the multiplex base-editing plasmids pMBEC*x* (Fig. 2B; see also next section and Supplementary Table 1). Here, multiple alternating gRNAs and Cas6 recognition sites can be introduced downstream of a P*trc* promoter through Golden Gate assembly. Note that the P*trc* promoter mediates constitutive gRNAs expression, and Cas6 is known to be catalytically active in multiple bacterial species[55,56]. Likewise, the σ70-dependent promoter driving the expression of *nCas9* fusions, P*rpsL*, is predicted to initiate transcription efficiently in several Gram-negative bacteria. In the following sections, we adopt a standard nomenclature to identify gene editing events, i.e., *gene*A*z**, where A indicates the amino acid encoded by a codon in position *z* that is converted into a *STOP* codon by cytidine editing. With this upgraded CBE toolbox at hand, the next step was to test and quantify key performance parameters of in vivo base-editing in Gram-negative species of interest.

**Enhanced efficiency and scope of base editing in *Pseudomonas* with the pBEC toolbox.** To test whether UGI influences base editing in *Pseudomonas* species, the efficiency of the CBE tool was

assessed in the presence or absence of UGI against several protospacers. The target 20-nt protospacers were selected such that a cytidine residue alternated with another base in the sequence, i.e., 5′-*NCN CNC NCN* C-3′ (Fig. 2D and Supplementary Table 2). Thus, seven protospacers were chosen, in which the cytidine base was distributed in all positions from 2 to 9, with all possible combinations of the preceding bases. In all cases, the matching protospacer sequences were retrieved by using the PatScanUI[57] platform, and protospacers annotated within an essential gene were discarded. These spacers were cloned either into vector pnCasPA-BEC (GmR, UGI−) or pBEC6 (GmR, UGI+) and the resulting plasmids were transformed into *P. putida* KT2440. Out of the seven protospacers selected to calibrate the tool; four targeted the coding strand, two targeted the non-coding strand, and one targeted an intergenic region. Upon editing, the target loci were amplified by PCR from individual clones and sequenced. In these experiments, base editing was considered successful if at least one cytidine was altered to thymidine (C·G → T·A) in the protospacer sequence. Due to the extended cultivation time allowed for the base editing process and subsequent plasmid curing, the genotypes segregated and mainly genotypic pure colonies were observed (Supplementary Fig. 1). In the rare case of double peaks in the chromatogram (mixed genotype), the ratio of the peak height was taken into account to calculate the relative abundance of each genotype. As a general trend, we observed that the editing efficiency decreased with an increasing steric size of adjacent bases in the protospacer. Editing worked very efficiently with pyrimidine-rich protospacers, adenine-rich ones were more difficult to modify, and guanine-rich protospacers could not be

edited at all (Fig. 2E). Along the same line, the base preceding (5′-end) the target cytidine residue has been reported to affect the editing efficiency[8]. We verified low editing efficiency with guanidine preceding the target base (Supplementary Fig. 2), but editing was completely omitted in the case of the guanine-rich protospacers—which indicates that the editing efficiency is affected not only by the preceding base(s) but also by subsequent residues at the 3′-end of the target. The trend observed in editing efficiency was T ≫ C > A with respect to the preceding base, and no editing when the 5′-base was a G (Fig. 2E). This trend was similar to the observations reported by Komor et al.[8] in experiments with BEs in vitro. Remarkably, the UGI fusion of the editor module boosted editing efficiency for both adenine-rich (from ca. 60 to 85%) and cytidine-rich spacers (from ca. 80 to 100%).

We also investigated if the fusion of UGI to the CBE modified the window of editing. CBEs have been reported to display a < 10-nt editing window in the PAM-distal region[58], regardless of the length of the linker between the CDA deaminase and $^{Sp}$Cas9$^{D10A}$. Indeed, UGI led to a significant broadening of the editing window for all spacers tested (Fig. 2F), and the extension of the editing window followed the same trend as observed for the overall efficiency (i.e., T > C > A > G, from broad to narrow). In all seven cases tested, the editing window was centered on position 6 of the protospacer. Based on these observations, the editable region of the BE system was assigned to 8 nt (i.e., positions 2–9 of the protospacer) with UGI and to 5 nt (i.e., positions 4–8 of the protospacer) without UGI in the PAM-distal sequence (Fig. 2F). Overall, the C residues in this stretch were converted into thymidine with frequencies between 40 and 100%—among the highest reported in the literature[14]. We thus concluded that UGI enhances the editing efficiency in *Pseudomonas* while broadening the window of editing. Interestingly, we did not observe any other mutation than C·G → T·A in our experiments (e.g., indels, C·G → G·C or C·G → A·T). While all these assays were conducted with single gRNAs to calibrate our CBE toolset, we wanted to explore the possibility of multiplexing base editing procedures in *Pseudomonas* as disclosed below.

**Highly-efficient, multiplex genome editing in Gram-negative bacteria with Cas6-bearing pMBEC plasmids**. Once the applicability of pBEC vectors was validated in *P. putida*, we expanded the scope of CBE-mediated genome engineering by multiplexed editing in different Gram-negative bacteria—both laboratory workhorses and non-traditional hosts. A set of pMBECx vectors was implemented to this end, where multiplexing is supported by the presence of *cas6f*, and the cloning of multiple gRNAs was simplified by fluorescence-assisted Golden Gate assembly and selection of positive constructs (Fig. 3A). Firstly, we introduced the monomeric, super-folder green fluorescent protein (msfGFP) gene[52] flanked by the two *Bsa*I recognition sites within the gRNA module. This feature facilitates screening and selection of pMBEC plasmids that contain the right spacer, since positive bacterial colonies become msfGFP⁻. Furthermore, several plasmid variants were generated, harboring different antibiotic resistance markers [i.e., kanamycin (Km, *x* = 2); streptomycin (Sm, *x* = 4); gentamicin (Gm, *x* = 6); and apramycin (Ap, *x* = 8)], in order to ease their use across bacterial species (Supplementary Table 1). The availability of these diverse selection markers helps circumventing natural resistances and potential incompatibilities, typical of non-traditional microbial hosts.

To test the pMBEC toolset for multiplex base-editing, we selected a gene target that enables rapid screening of positive, base-edited bacterial clones. *P. putida* KT2440 can grow on nicotinic acid (NA)[21]. The degradation pathway, encoded in the

*nic* gene cluster, involves NA hydroxylation to 6-hydroxynicotinic acid, reduction to 2,5-dihydroxypyridine, and deoxygenation to *N*-formylmaleamic acid, which is further converted into fumarate[59] (Fig. 3B). Interrupting the metabolic route by eliminating NicX (2,5-dihydroxypyridine 5,6-dioxygenase) leads to the accumulation of 2,5-dihydroxypyridine, a dark green-colored compound that forms brownish polymers upon autoxidation[29]. Hence, editing *nicX* (PP_3945) was chosen to optimize our genome engineering protocol as it allows for direct identification of mutants when cultivated in the presence of NA (Fig. 3C). Next, a RNA cassette was designed so that processing by Cas6 produces five individual gRNAs. The first gRNA in the cassette, targeting *nicX* (and leading to the *nicX*$^{W187*}$ functional knock-out), was used to assess the editing efficiency. The assembly was cloned in plasmid pMBEC6 (Gm$^R$, UGI$^+$, Cas6$^+$), and transformed into strain KT2440. Upon a 24 h incubation, cells were plated onto LB medium agar containing sucrose, and 96 individual clones were grown in liquid LB medium with 5 mM NA in microtiter plate cultures (Fig. 3C). Under these conditions, almost 100% of the clones displayed a strongly-colored phenotype and contained the *nicX*$^{W187*}$ edit that leads to a premature *STOP* codon in the gene.

As this initial test was successful, we then changed the gRNAs order to investigate the dependence of editing efficiency on the gRNA position in the cassette. The other four targets were *benA*, *gclR*, *glpR*, and *nfxB*; chosen because the cognate loci are scattered across the bacterial chromosome. All the first four positions mediated an editing efficiency >96% as scored by the NicX⁻ phenotype in NA-containing LB medium (Fig. 3D). When the *nicX* spacer was placed at the last (fifth) position in the gRNA cassette, however, the editing efficiency dropped to 14%. The low efficacy could potentially stem from slow kinetics of the system, and we extended the editing period to 48 h to circumvent this poor editing performance. Hence, the bacterial culture was diluted at 24 h and incubated again under the same conditions to allow for more cell doublings. This operation resulted in >5-fold increase in the editing efficiency when the *nicX* gRNA was placed at the fifth position in the cassette (Fig. 3D), and did not significantly affect editing when the spacer was located in any of the first four positions. Nevertheless, extending incubation intervals prolongs the working time, and we explored alternative strategies to enhance the editing efficiency mediated by the last gRNA.

We hypothesized that the transcription of distal gRNAs in the multiplex construct might be lower than those closer to the P$_{trc}$ promoter, affecting the editing efficiency by unbalanced availability of mature gRNAs. To test this hypothesis, we constructed cassettes with varying numbers of gRNAs, with the *nicX* spacer always placed in the last position. These constructs were tested as explained above, and the last gRNA consistently mediated low editing efficiency independently of the cassette length (Supplementary Fig. 3). Since the positional effect is independent of the RNA length, we turned our attention to the fact that the last gRNA does not carry a Cas6 recognition site. Instead, it carries a short terminator sequence composed of seven consecutive uracil residues. By altering this terminator region into a Cas6 recognition site, we wanted to investigate its potential role on editing efficiency. Introducing a Cas6 recognition site after the *nicX* gRNA in the last position of the cassette increased base editing efficiencies to 93% upon a 24 h incubation (Fig. 3D). Moreover, when the incubation period was extended to 48 h, the editing efficiency was as high (>98%) as for any other gRNA position in the cassette. The exact reason of this positive effect of Cas6 remains unknown, yet it seems plausible that binding of the endoribonuclease either (i) protects the gRNA against degradation (thereby increasing the effective amount of gRNA available

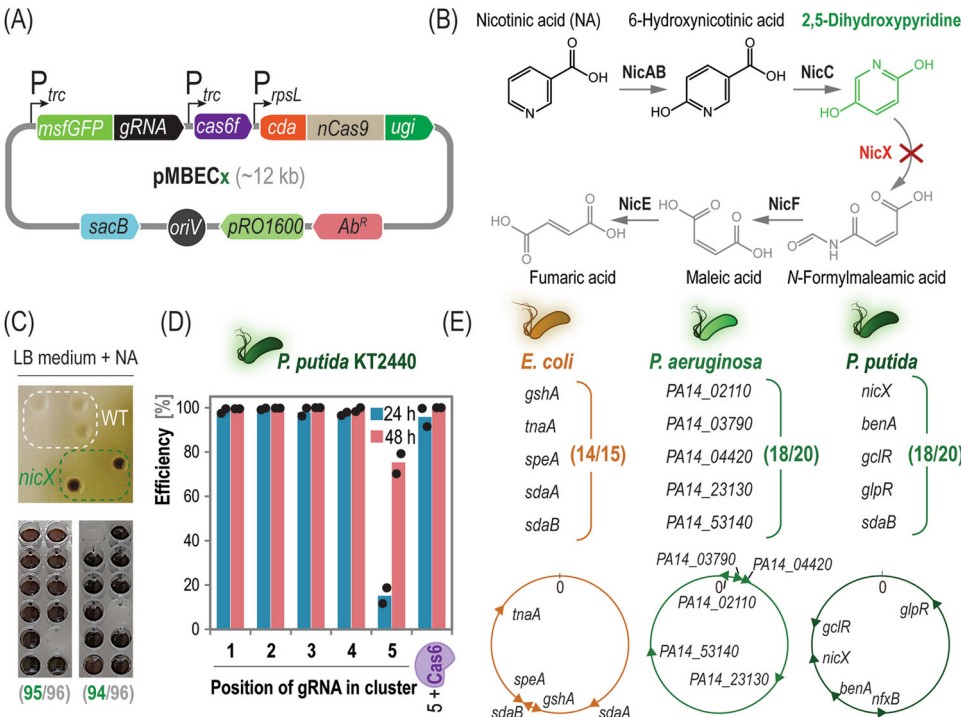

**Fig. 3 Testing multiplex base editing across bacterial species. A** Architecture of pMBEC base editing plasmids containing the monomeric, superfolder GFP gene (*msfGFP*) in place of the spacer in the guide RNA (*gRNA*).Both, *gRNA* and *cys6f* are transcribed from the *trc* promoter ($P_{trc}$). The construct encoding a fusion protein between the cytidine deaminase (CDA), the nicking Cas9 (nCas9) and the UNG inhibitor (UGI) is transcribed with a *rpsL* promoter ($P_{rpsL}$) Note that **x** is a placeholder identifying antibiotic resistances ($Ab^R$). **B** Nicotinic acid (*NA*) degradation in *P. putida* KT2440. Loss-of-function mutations in *nicX* lead to 2,5-dihydroxypyridine accumulation; spontaneous oxidation and polymerization of this intermediate yield a green coloration of NicX⁻ colonies. **C** High-throughput phenotypic screening of wild-type (WT) *P. putida* and *nicX*^W187* mutants on lysogeny broth (LB) and M9 medium cultures supplemented with NA. The number of *nicX* mutants and total colonies tested are indicated in each case. **D** Editing efficiency of *nicX*^W187* at each position of a five gRNA cluster. An additional Cas6-recognition site was inserted behind the last gRNA (5+Cas6) to boost efficiency. Mean values of two independent biological experiments are presented; dots represent individual data per experiment. **E** Multiplex base editing in *E. coli*, *P. aeruginosa* and *P. putida*, simultaneously targeting five genes across genome loci. The number of successfully edited variants is indicated for each experiment together with physical chromosome maps. Source data underlying panel **D** are provided as a Source Data file.

for editing), (ii) promotes interactions between nCas9 and the gRNA, or (III) the 3′-cyclic phosphate group left by the activity of Cas6 delays gRNA degradation[60]. Regardless, and since the addition of a Cas6 recognition site mediated an increased editing efficiency, we routinely included this feature in subsequent designs.

The versatility of the pMBEC toolset was put to test by attempting base editing in other Gram-negative bacterial species besides *P. putida* KT2440. To this end, five targets were chosen in *E. coli* and *P. aeruginosa*, covering loci scattered across the bacterial chromosomes in both orientations (Fig. 3E). In *E. coli* BL21, a platform strain used for protein overproduction[61], *gshA* (γ-glutamate-cysteine ligase), *tnaA* (tryptophan indole-lyase), *speA* (arginine decarboxylase), *sdaA* (L-serine deaminase I), and *sdaB* (L-serine deaminase II) were simultaneously targeted (Supplementary Table 2). Upon a 24 h incubation of *E. coli* BL21 transformed with a derivative of plasmid pMBEC6 that contains the five gRNAs, two out of three randomly picked clones had all targets edited at the desired positions, while only one of them displayed four out of five edits in place. Likewise, five targets were simultaneously edited in *P. aeruginosa* PA14 (Supplementary Table 2), an opportunistic pathogen[17], i.e., *PA14_02110* (a diguanylate cyclase), *PA14_03790* and *PA14_23130* (sensory box GGDEF domain-containing proteins), and *PA14_04420* and *PA14_53140* (annotated as hypothetical proteins). In this case, two out of four randomly selected clones were completely edited, while two clones had four out of five targets successfully modified.

Similarly, four clones of *P. putida*, targeted in *gclR*, *nicX*, *benA*, *nfxB*, and *glpR*, were analyzed. Three of them totally edited, while one clone was only edited in three protospacers.

Base-editors are known to mediate the emergence of off-target mutations, i.e., mutations outside the protospacer. These modifications can be caused either by the activity of the deaminase on exposed single-stranded DNA independently of protospacer recognition (e.g., during DNA replication), or by inhibition of the DNA repair machinery through UGI. In order to investigate these potential effects in our toolset, *P. putida* KT2440 was transformed either with the empty pSEVA631 vector (included as a control to assess the basal mutation frequency), a base-editing plasmid without a spacer (plasmid pBEC), or a base-editing plasmid carrying the C1 spacer (Supplementary Table 2), with or without UGI (plasmid pnCas9-BEC-C). All strains were treated according to the base-editing protocol and subsequent plasmid curing procedures described above. While the strain carrying the empty vector displayed 8 spontaneous mutations, the base-editing plasmid with and without spacer resulted in the emergence of 14 and 18 mutations, respectively (Supplementary Table 4 and Supplementary Data 2). Additionally, co-expressing *ugi* led to 33 mutations. Therefore, expression of a Cas9-deaminase fusion leads to a ca. two-fold increase in the mutation rate independently of the presence of a spacer, while the activity of UGI boosts mutation rates by four-fold. These figures of unintended modifications are comparable to other genome-modification tools based on BEs[62,63] and appear to be acceptable

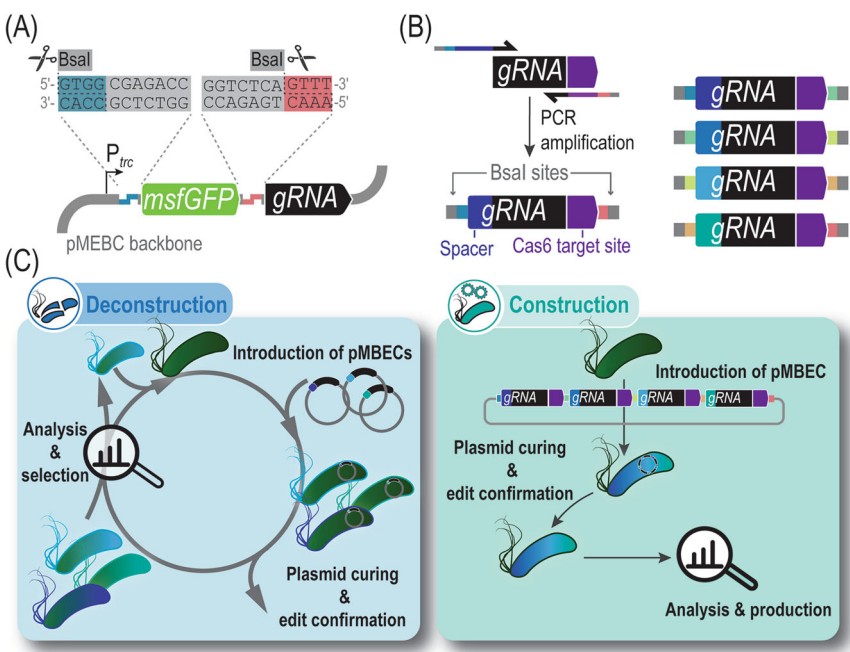

**Fig. 4 Workflow for base editing-mediated (de)construction of complex bacterial phenotypes. A** To facilitate cloning procedures, a fluorescent protein marker was added to the pMBEC backbone, facilitating rapid counterselection of template vectors. The base editing-plasmid contains a constitutively-expressed *msfGFP* module flanked by BsaI recognition sites. This marker is replaced by either one spacer sequence or multiple gRNAs through Golden Gate assembly. **B** Generation of gRNAs through PCR for multiplexed genome editing. The Cas9 handle is amplified along with a Cas6 recognition sequence from a template vector, while the specific spacer sequence and Golden Gate-flanking motifs are introduced in the oligonucleotide sequences. The resulting gRNAs harbor unique BsaI sites that can be used to compose multiplex arrays by Golden Gate cloning. **C** pMBECs vectors enable deconstruction and construction of complex phenotypes by multiple genome editing. SacB-mediated curing of these plasmids facilitates multiple editing cycles upon confirmation and analysis of the resulting bacterial phenotypes.

for most practical applications (e.g., metabolic engineering). Taken together, these results expose the versatility of multiplex genome editing afforded by the pMBEC toolset, and the next step in our study was to optimize an automated workflow for editing-plasmid construction.

**Optimized protocol for multiplex editing plasmid construction by standardized part assembly and fluorescence-assisted screening.** User-friendliness and robustness make synthetic biology tools easy to implement by the broad research community. Thus, we aimed not only at boosting the performance of the base-editing tool in *Pseudomonas*, but also at developing a protocol for one-step assembly of pMBEC plasmids equipped with multiple gRNAs (Fig. 4). Out of the many different cloning techniques adopted thus far, Golden Gate assembly is among the most widely implemented[64]. We selected this methodology because the assembly efficiency is not impacted by the repetitive nature of the gRNAs, and since amplification of the relatively large vector backbone (which can lead to undesired mutations during PCR) is not needed. As indicated above, the presence of *msfGFP* in the multiple cloning site of the pMBEC vectors greatly facilitated the screening of plasmids carrying gRNAs cloned by *Bsa*I digestion and ligation (Fig. 4A). However, we observed that the rate of successful cloning events was rather low for constructs composed of eight or more gRNAs, and often made it necessary to redesign the parts needed for assembly.

The fidelity of Golden Gate assembly was shown to rely heavily on the identity and combination of overhangs[65]. Our original Golden Gate strategy utilized 4 bp-long restriction overhangs at the 5′-end of the spacer sequence, rendering the overhang specific for each spacer. A data-optimized design by overhang standardization[66] was implemented towards high-fidelity Golden

Gate reactions and reproducible results independently of the spacer complexity and sequence. To this end, we placed a 4-bp adapter in the 3′ terminus of the Cas6 target site (Fig. 4B), which is not expected to interfere with Cas6 as it lies outside the essential recognition motif[53]. This strategy significantly reduces cloning costs because a single oligonucleotide primer is ordered for each spacer. The complementary, reverse primer (5′-ATC GAG GTC TCC NNN NTT AGC TGC CTA TAC GGC AGT-3′, (Supplementary Table 3) is specific for the position of the gRNA in the RNA cassette, depending on the nature of the 4 bp linker (identified as NNNN in the oligonucleotide sequence), and can be reused as needed. Indeed, using standardized overhangs reproducibly enabled assembly of up to 12 gRNAs in different RNA cassettes with >60% correctness—tested by scoring 4–10 randomly-picked colonies after transformation of chemically-competent *E. coli* with the Golden Gate reaction. An optimized protocol for Golden Gate assembly of multiple gRNAs is presented in the Supplementary Methods. To further simplify this procedure, we also developed a script for the one-step design of oligonucleotides (Supplementary Data 3), where the only input needed is the sequence of the protospacer to be targeted. By combining these individual optimization steps, we set to apply the upgraded CBE toolbox for design and (de)construction of complex bacterial phenotypes. In this context, we define *deconstruction* as the top-down engineering of genome modifications, often in a cyclic fashion; whereas *construction* is presented as the one-step multiplex editing of gene targets that lead to the phenotype of interest (Fig. 4C). Two application examples are presented in the next sections.

**One-step construction of an engineered *P. putida* strain optimized for PCA production.** PCA, a hub metabolite in the

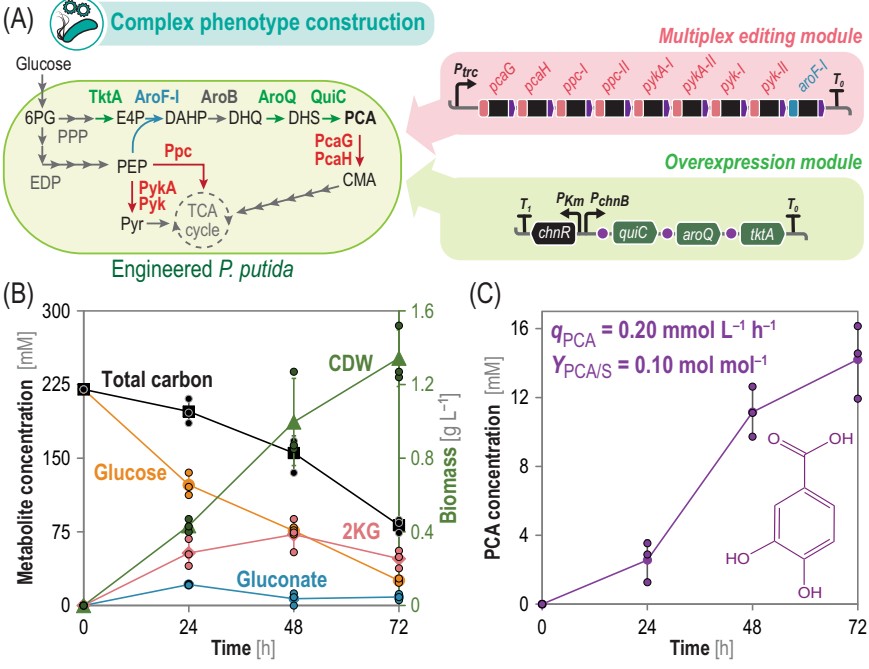

**Fig. 5 One-step engineering of *P. putida* for glucose-dependent production of PCA. A** Simplified metabolic map indicating key reactions eliminated or modified towards protocatechuate (*PCA*) overproduction. Editing *pcaG* (PP_4655), *pcaH* (PP_4656), *ppc* (PP_1505), *pykA* (PP_1362), and *pyk* (PP_4301) blocked the cognate biochemical reactions, while *aroF-I* (PP_2324) was modified to remove feedback inhibition on phospho-2-dehydro-3-deoxyheptonate aldolase. All of these modifications are encoded in a single, multiplex editing module, containing nine gRNAs. Additionally, *quiC*, *aroQ* and *tktA* were overexpressed as a synthetic operon under ChnR/$P_{chnB}$ transcriptional regulation. Genetic elements not drawn to scale. *6PG*, 6-phosphogluconate; *PEP*, phosphoenolpyruvate; *Pyr*, pyruvate; *E4P*, erythrose 4-phosphate; *DAHP*, 3-deoxy-D-arabinoheptulosonate 7-phosphate; *DHQ*, 3-dehydroquinate; *DHS*, 3-dehydroshikimate; *CMA*, β-carboxy-*cis,cis*-muconate; *EDP*, Entner-Doudoroff pathway; *PPP*, pentose phosphate pathway; *TCA cycle*, tricarboxylic acid cycle. **B** Growth and substrate consumption and **C** product formation in shaken-flask fermentations of the edited strain. *CDW*; cell dry weight; *2KG*, 2-ketogluconate; $q_{PCA}$, PCA volumetric productivity; $Y_{PCA/S}$, PCA yield on substrate. Data represent mean values ± standard deviation of three independent biological experiments. Source data underlying panels **B**, **C** are provided as a Source Data file.

catabolism of aromatic compounds by environmental bacteria[67], attracted interest as building block for flavor and fragrance production[68]. De novo biosynthesis of PCA from sugar feedstock calls for an labor-intense microbial engineering program, involving gene deletion, overexpression, and pathway fine-tuning[69]. *P. putida* KT2440 is endowed with all genes needed for biosynthesis of PCA through the shikimate route;[21] however, the pathway naturally carries a low flux and PCA is degraded by the native PcaGH (PCA 3,4-dioxygenase) enzyme[70]. Therefore, we set out to engineer a *Pseudomonas* strain for PCA production based on endogenous functions—rather than extensively expressing heterologous genes. Firstly, we constructed plasmid pS2311·PCA (Supplementary Table 1), which carries *quiC* (3-dehydroshikimate dehydratase), *aroQ-I* (3-dehydroquinate dehydratase), and *tktA* (transketolase) under control of the cyclohexanone-inducible ChnR/$P_{chnB}$ system[71] (Fig. 5A). This overexpression module was designed such that QuiC and AroQ, identified to mediate rate-limiting reactions[72,73], are no longer a bottleneck towards PCA biosynthesis. Likewise, *tktA* overexpression should increase erythrose 4-phosphate availability to feed the shikimate pathway[74,75]. Overexpression of these genes alone, however, did not lead to any detectable PCA production—probably caused by the degradation of the aromatic or by draining of key intermediates through competing pathways.

To prevent product degradation, we established functional *pcaG* and *pcaH* knock-outs through base-editing at positions Q81 and W16, respectively. Both genes are mutually essential for PCA breakdown (i.e., they encode the α and β subunits of the dioxygenase), and a single *STOP* codon per ORF should limit the emergence of revertants. Next, in order to boost phosphoenolpyruvate (PEP) availability, the second substrate for the shikimate pathway[75,76], we targeted inactivation of *ppc*, *pykA*, and *pyk*. Spacer sequences were designed to introduce double *STOP* codons at positions Q74 and W247 of *ppc*, Q24 and W217 of *pykA*, and W52 and Q59 of *pyk*. Finally, we also manipulated post-translational regulation of metabolic nodes relevant for PCA biosynthesis. AroF, 3-deoxy-D-arabinoheptulosonate 7-phosphate synthase, is the first committed step of the shikimate route and is subjected to feedback inhibition by aromatic amino acids[77]. To release AroF from this inhibitory mechanism, and to increase the overall flux through the shikimate route, we intended to introduce a D159N mutation (analog to D146N of *aroG* in *E. coli*[78]) in PP_2324 (*aroF-I*) by base-editing. Hence, a derivative of vector pMBEC carrying gRNAs needed to edit all nine targets were assembled (Fig. 5A) and transformed into wild-type *P. putida* KT2440. After curing the plasmid by two passages on sucrose-containing LB medium, colony PCR of randomly picked-isolates and sequencing of DNA fragments containing the targets confirmed successful editing of all nine genes. Whole genome sequencing further corroborated the editing of the intended targets, with only 23 additional, off-target mutations (Supplementary Table 4 and Supplementary Data 2). Importantly, none of these mutations leads to the unexpected interruption of ORFs. The very low number of off-target modifications in comparison to the envisioned mutations is probably due to the simultaneous editing of the targets followed by quick curing of the base-editing plasmid.

This genome-edited *P. putida chassis* was transformed with the gene overexpression module, and the resulting strain (*P. putida*

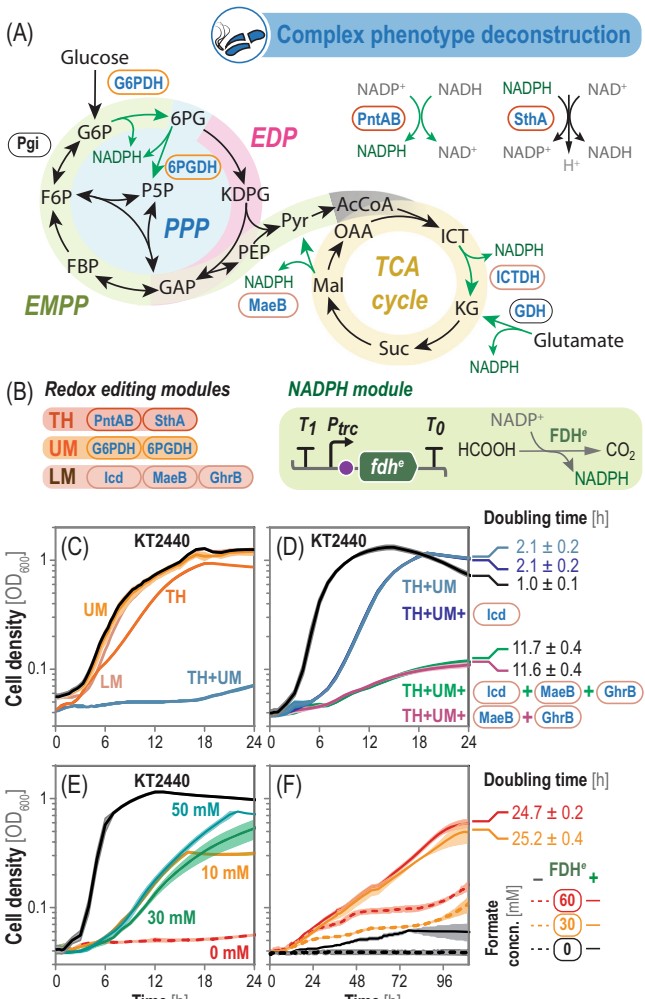

**Fig. 6 Modular deconstruction of the NADPH metabolism in *P. putida*.**
**A** Simplified scheme of central metabolism in *P. putida*. Four core reactions deliver NADPH, i.e., glucose 6-phosphate dehydrogenase (G6PDH), 6-phosphogluconate dehydrogenase (6PGDH), malic enzyme (MaeB), and isocitrate dehydrogenase (ICTDH). The PntAB and SthA transhydrogenases catalyze electron transfer between nicotinamide nucleotides to buffer cofactor imbalances, while the reversible glutamate dehydrogenase (GDH) can supply NADPH when glutamate is fed. **B** Three redox editing modules were defined as to constrain NADPH supply either through transhydrogenation (*TH*), upper metabolism (*UM*; G6PDH and 6PGDH), or lower metabolism (*LM*; Icd, MaeB, and GhrB). NADPH formation within the UM when cells are grown on gluconeogenic substrates was prevented by interrupting the two phosphoglucoisomerase (Pgi) genes. In some experiments, a NADPH-dependent formate dehydrogenase (FDH^e) gene was constitutively expressed in a NADPH-depleted strains to restore redox balance. **C** *P. putida* KT2440 and genome-edited strains with a combination of the UM, LM, and TH modules were grown in de Bont minimal medium with 30 mM pyruvate. **D** The TH + UG strain was further engineered by knocking out *icd*, *ghrB*, and *maeB*, or their combination (i.e., the NADPH-depleted *P. putida* strain). These strains were grown in de Bont minimal medium with 30 mM pyruvate and 50 mM α-ketoglutarate. **E** Growth profile of the NADPH-depleted *P. putida* strain in de Bont minimal medium with 30 mM pyruvate and varying glutamate concentrations as indicated in the graphic. **D** Growth profile of the NADPH-depleted *P. putida* strain, carrying either a NADP^+-dependent formate dehydrogenase (solid line) or an empty plasmid (dashed line), in de Bont minimal medium with 30 mM pyruvate and varying formate concentrations (concn.). Data represent mean values of three independent biological experiments, and doubling times ± standard deviations were derived from the optical density at 600 nm (OD_{600}) measurements. Source data underlying panels **C**–**F** are provided as a Source Data file.

PCA, Supplementary Table 1) was grown in shaken-flask fermentations with de Bont minimal medium using 4% (w/v) glucose as the sole carbon substrate. During the 72 h experiment, *P. putida* PCA utilized half of the glucose for growth and product formation, while the other half of the sugar was converted to the gluconate and 2-ketogluconate by-products (Fig. 5B), a feature typically observed in sugar-dependent growth of *P. putida*[79]. Nevertheless, *P. putida* PCA actively produced the target aromatic compound under these conditions, with the PCA concentration reaching 14.2 ± 2.1 mM by the end of the fermentation with a volumetric productivity $q_{PCA} = 0.20 ± 0.03$ mmol L$^{-1}$ h$^{-1}$ (Fig. 5C). These PCA titers and $q_{PCA}$ values are among the highest reported in strains constructed by the existing genome engineering methods[73]. Moreover, in a best-case scenario, it would take up to 4 weeks to implement these modifications in any given *Pseudomonas* strain[33]. The one-step construction of such a complex production strain prompted the question of whether an equally intricate metabolic network can be depieced using the CBE toolbox.

**Deconstructing the redox metabolism of *P. putida* towards a synthetic 'NADPH-auxotrophic' strain.** *P. putida* is naturally endowed with a high NADPH turnover capacity, attributed to a cyclic glycolysis architecture and the presence of cofactor-flexible dehydrogenases[79–81]. A major sink of NADPH is thioredoxin and glutathione reduction to keep a reduced milieu in the cytoplasm[82]. NADPH turnover in the upper metabolism can be adjusted either by (i) increasing fluxes through glucokinase and

glucose 6-phosphate dehydrogenase (G6PDH) or (ii) cycling through the pentose phosphate pathway, starting with G6PDH and 6-phosphogluconate dehydrogenase (Gnd)[79]. However, the prevailing mechanism supporting high NADPH turnover remains unknown. The redox metabolism of strain KT2440 also encompasses five major dehydrogenases (Fig. 6A), known to accept NADP$^+$ as a cofactor[83,84]. Furthermore, *P. putida* harbors the (membrane-bound) PntAB and (soluble) SthA transhydrogenases, which transfer electrons between the redox pairs NADH/NADP$^+$ and NADPH/NAD$^+$, respectively. On this background, we investigated the physiological relevance and potential essentiality of NADPH-producing reaction(s) by a top-down approach, where the CBE tool was implemented to introduce functional knock-outs in the cognate genes. Most NADP$^+$-dependent dehydrogenases are deemed important (or even essential) for growth of *P. putida* on glucose, hence a single-step gene-editing approach (as implemented for engineering the PCA production strain) was considered impractical. Instead, we focused on a modular, iterative gene-editing strategy, whereby key metabolic nodes are sequentially blocked or isolated to limit NADPH formation (Fig. 6B). Three different editing modules were designed to target NADPH production (i) in the upper metabolism (*UM*, i.e., glucose to pyruvate), (ii) in the lower metabolism (*LM*, i.e., downwards pyruvate), and (iii) through pyridine nucleotide transhydrogenation (*TH*, i.e., PntAB and SthA).

The deconstruction strategy started by blocking glucose 6-phosphate isomerase (Pgi, encoded by *pgi*-I and *pgi*-II). In this way, the four UM reactions that yield NADPH, i.e., G6PDHs (encoded by *zwfA*, *zwfB*, and *zwf*) and Gnd (encoded by *gntZ*), can be decoupled from the rest of the metabolism during gluconeogenic growth (i.e., NADPH formation can be limited in

such a strain by feeding non-sugar substrates). Also, this intervention leaves NADPH production unrestrained under a glycolytic regime, which can be adopted as a 'relaxing' growth condition. Within LM, NADPH regeneration is mediated by isocitrate dehydrogenase (ICTDH) in the tricarboxylic acid (TCA) cycle, and the anaplerotic malic enzyme (MaeB). The two ICTDH isoforms of strain KT2440, annotated as NADP$^+$-dependent enzymes, are encoded by the mutually essential genes *idh* (monomeric) and *icd* (dimeric)[85,86]. Additionally, GhrB (a 2-ketoaldonate/hydroxypyruvate/glyoxylate reductase) was identified as a NADP$^+$-dependent dehydrogenase with promiscuous activity on LM intermediates by bioinformatic analysis[21]. Thus, *ghrB* was targeted along with *icd* and *maeB*. Besides wild-type strain KT2440, a *P. putida* strain harboring clean Δ*sthA* deletions[87] and edited in *pntAB* (i.e., targets within the TH module) was used for introducing these modifications. Separate pMBEC plasmids, carrying the UM, LM, and TH editing modules (Fig. 6B), were constructed and delivered into the host strains as indicated above. Upon sucrose-mediated curing of the plasmids, we confirmed that the intended mutations have been introduced in the genes of interest by colony PCR and sequencing of the resulting amplicons. As a further confirmation of the engineered genotype, the genome of the final NADPH-depleted strain, carrying base modifications in all three functional modules, was fully sequenced (Supplementary Table 4 and Supplementary Data 2). We could confirm the presence of all intended mutations in the UM, LM, and TH modules. Furthermore, the strain accumulated 94 off-target mutations, which could be accounted for by the extended cultivation time (as this engineered strain grows very slowly, see below) combined with a highly-oxidative intracellular environment due to low NADPH levels[88]. The next step was a quantitative physiology characterization of all modified *P. putida* strains in de Bont minimal medium under either glycolytic (glucose) or gluconeogenic (pyruvate) conditions.

Editing of genes within the UM and LM modules did not result on any substantial growth reduction on the gluconeogenic substrate pyruvate (Fig. 6C). This pattern is probably due to the flexibility and redundancy of NADPH metabolism in *P. putida*, as the deficiency of one metabolic block can be counteracted with the NADPH supplied by other activities. As predicted, glucose-dependent growth was not affected. Eliminating the PntAB and SthA transhydrogenases, in contrast, increased the doubling time (DT) of the resulting strain by 56%, and combining the TH and UM modules altogether abolished pyruvate-dependent growth (Fig. 6C). Since the TH + UM strain could not thrive under a gluconeogenic regime, we reasoned that the NADPH supply within LM was not enough to support growth. Using pyruvate as the carbon feedstock, for instance, could shift the equilibrium of MaeB towards the reverse reaction. When the experiment was repeated in the presence of pyruvate and α-ketoglutarate (KG, an intermediate of the TCA cycle), the cell densities reached by the TH + UM strain were similar to that of wild-type *P. putida* (Fig. 6D). Under these conditions, the DT of the TH + UM strain was double that of the wild-type strain, indicating that NADPH formation can be modulated at the level of carbon substrate availability probably at the level of MaeB.

To test this possibility, we eliminated the remaining NADPH-producing reactions on the TH + UM background to identify key enzymes involved in redox balance. While editing *icd* did not alter growth patterns, inactivation of *maeB* and *ghrB* significantly impacted the strain physiology (Fig. 6D), extending DTs by >11 h. Since the function of GhrB should not be altered by addition of KG to the medium, MaeB emerges as the most likely NADPH-supplying reaction. Indeed, *icd* editing in this strain did not lead to a further reduction in growth kinetics, and blocking

*maeB* individually led to the same growth phenotype as editing both *maeB* and *ghrB*. The redundant ICTDH activity, carried by Icd and Idh, explains why individual deletions do not impact redox balance significantly, and the mutual essentiality of *icd* and *idh*[85] was evidenced by the fact that we could not edit both genes simultaneously. Regardless, these experiments not only allowed to depiece the complex redox metabolism of a model bacterial species, but they also led to the construction of a NADPH-deficient *chassis* edited in all three modules (i.e., *P. putida* Δ*sthA* *pntA*$^{W238*}$ *pntB*$^{Q117*}$ *pgi*-I$^{Q129*,W229*}$ *pgi*-II$^{Q129*}$ *ghrB*$^{W124*}$ *maeB*$^{Q314*}$ *icd*$^{Q78*}$, Supplementary Table 1) that can be used as a selection strain for NADPH-producing reactions—both endogenous and heterologous.

To explore the potential 'awakening' of silent NADPH-producing reactions in the metabolic network, the synthetic 'NADPH-auxotrophic' strain was grown under restrictive conditions (i.e., de Bont medium amended with 30 mM pyruvate) and varying glutamate concentrations (Fig. 6E). We reasoned that the highly-reversible and metabolically silent in sugar-based cultivations L-glutamate dehydrogenase (GDH)[89] could oxidize the amino acid to KG under these conditions, thereby generating additional NADPH. In contrast to KG addition, glutamate restored growth of the NADPH-deficient strain (Fig. 6E). Interestingly, this complementation effect was verified even at low glutamate concentrations, which indicates that NADPH supply by GDH (and not the extra carbon and nitrogen provided by glutamate) is mainly responsible for rescuing the growth defect of the NADPH-depleted *P. putida* strain.

Furthermore, we tested if the synthetic NADPH deficiency in the fully genome edited-strain could be countervailed by the action of heterologous dehydrogenases. Thus, we constructed a NADPH module consisting of an engineered formate dehydrogenase from *Pseudomonas* sp. strain 101 evolved[90] to steer its cofactor specificity towards NADP$^+$ (FDH$^e$), thereby providing the reduced nucleotide while formate is oxidized to $CO_2$ (Fig. 6B). The NADPH-deficient strain was transformed with either plasmid pFDH, harboring the engineered *fdh*$^e$ under transcriptional control of the constitutive P$_{trc}$ promoter (Supplementary Table 1), or the empty vector. The cells were incubated under redox-restrictive conditions (i.e., de Bont medium with 30 mM pyruvate) and added with formate as the FDH$^e$ substrate. While no growth was detected in the absence of formate even upon prolonged cultivation, FDH$^e$ effectively rescued the phenotype of the NADPH-auxotrophic *P. putida* incubated in the presence of either 30 or 60 mM formate (with DT values around 30 h, Fig. 6F). No mutants that could escape the NADPH-based selection scheme seem to have arisen—highlighting that the insertion of two premature STOP codons is sufficient to prevent the emergence of revertants in the time scale of typical laboratory experiments. Interestingly, weak growth of the strains carrying the empty vector was also verified in the presence of the C1 additive after 48 h. While this growth phenotype was not as clear as that of strains carrying FDH$^e$, it could reflect an 'awakening' phenomenon of native formate dehydrogenases of strain KT2440 that are usually silent[91]—and this redox-demanding culture conditions could have triggered such activities. These observations highlight the value of the top-down approach afforded by the CBE toolbox towards understanding and manipulating the redox metabolism of biotechnologically-relevant bacteria.

## Discussion

The cytidine base-editor developed here is a fast and robust genome engineering tool for Gram-negative bacteria that provides access to microbial engineering programs that were impossible thus far. The versatility of this CBE toolset is

particularly important when using non-traditional microorganisms (including natural bacterial isolates) as hosts, for which dedicated genome engineering techniques may be underdeveloped or altogether lacking. We demonstrated how complex phenotypes can be built in a single-step (i.e., PCA production from sugars) and how consecutive base-editing cycles can be leveraged for disentangling intricate metabolic networks (i.e., redox homeostasis in a NADPH-depleted *P. putida* strain). A key to boost the efficiency of the CBE toolset was the implementation of UGI, which also broadened the editing window. Likewise, incorporating Cas6 did not only allow for multiplexing of target loci, but also increased editing efficiency to levels among the highest reported for any genome editing tool thus far. Importantly, our CBE technology enables target, multiplex modifications in a variety of bacteria, including non-traditional microbial hosts, e.g., *Pseudomonas* species, where tools commonly applied to model bacteria, e.g., λ-Red recombineering, display very low efficiencies[39,92].

As the phenotypic alterations introduced by base-editing should be stable under laboratory time scales, we asked the question of how probable reversions of engineered *STOP* codons are. The mutation frequency is $z \approx 1 \times 10^{-9}$ per base per generation per bacterial genome[93]. Considering that around half of the *STOP* codons are TAA, the probability $P_1$ of converting them into an amino acid-coding codon is $P_1 = (1 + 2/3 + 2/3) \times z$. Other *STOP* codons, either TAG or TGA, have a probability of reverting $P_2 = (1 + 1 + 2/3) \times z$. Thus, the average reversion probability is $2.5 \times z = 2.5 \times 10^{-9}$ per base per generation per genome. For a *double STOP* codon insertion in the same ORF, this probability becomes $6.25 \times z^2 = 6.25 \times 10^{-18}$ per base per generation per genome. Accordingly, the chances that a double-edited ORF reverts to the wild-type sequence are very small, even after many doublings. We also note that, as it is the case with currently-available genome engineering techniques, base-editing procedures can lead to the introduction of unwanted mutations. These off-target effects have been explored by whole genome sequencing of selected strains created in this work. Indeed, off-target modifications above the background mutation level could be detected in all base-edited *P. putida* strains. These results highlight how multiplexing base-editing protocols offers an additional advantage over sequential editing. The presence of multiple gRNAs does not increase the mutation rate as compared to that detected in the presence of a single gRNA. That the number of off-target effects is not higher with multiple gRNAs also indicates that false protospacer recognition is not a major source of unwanted mutations. Furthermore, as the time of base-editing is reduced in comparison with serial modifications (involving several individual CBE plasmids), the number of off-target mutations is greatly reduced. Regardless of these observations, the protocols used for base-editing can be tailored for any given bacterial host by decreasing the time that the CBE plasmid is kept in the bacterium—thereby decreasing the emergence of off-target modifications at the expense of mutation efficiency.

In conclusion, we expect that the DNA engineering system and protocols reported in this study will be widely adopted by genetic and metabolic engineers not only for *Pseudomonas*, but also other organisms. Due to its modular character and the simple layout (enabling module swapping and composability), this toolset is likely to advance our understanding of fundamental aspects of microbial physiology and to increase capabilities for complex microbial engineering for biotechnological purposes.

## Methods

**Bacterial strains, plasmids, and culture conditions**. Bacterial strains and plasmids employed in this study are listed in Supplementary Table 1. *E. coli* and *P. aeruginosa* were routinely incubated at 37 °C, while *P. putida* was grown at 30 °C.

For standard applications, cloning procedures and during genome engineering manipulations, bacteria were grown in LB medium (10 g L⁻¹ tryptone, 5 g L⁻¹ yeast extract, and 10 g L⁻¹ NaCl, pH = 7.0). NA and PCA were purchased from Sigma-Aldrich Co. (St. Louis, MO, USA) and used at the concentrations indicated in the text. When isolating *E. coli* clones, antibiotics were added to the culture media at the following concentrations: Gm, 10 µg mL⁻¹; ampicillin, 100 µg mL⁻¹; Km, 50 µg mL⁻¹; apramycin, 25 µg mL⁻¹ and streptomycin 50 µg mL⁻¹. The same concentrations were used in *P. putida* and *P. aeruginosa* cultivations, except for streptomycin (100 µg mL⁻¹ in both cases) or Gm (30 µg mL⁻¹ for *P. aeruginosa*).

For PCA production experiments, 35 mL of de Bont minimal medium[94], supplemented with 4% (w/v) glucose, Km and 1 mM cyclohexanone, was inoculated with 0.3 mL of an overnight culture of the relevant engineered *P. putida* strain previously grown in LB medium. Cultures were incubated in 250 mL baffled flasks with rotary agitation at 200 rpm (MaxQ™ 8000 incubator; ThermoFisher Scientific, Waltham, MA, USA). Samples were periodically withdrawn for analytical quantifications as indicated. The physiological characterization of NADPH-depleted *P. putida* strains was performed in de Bont minimal medium supplemented with either 30 mM pyruvate, 30 mM pyruvate and 50 mM α-ketoglutarate, or 30 mM pyruvate and glutamate or formate at the concentrations specified in the text. Cultures (200 µL) were incubated in 96-well microtiter plates, and optical densities were recorded in a microplate reader (Elx808™, BioTek Instruments; Winooski, VT, USA). Growth parameters were obtained from OD-*versus*-time plots.

**General DNA manipulations and plasmid construction**. Spacers and oligonucleotides are listed in Supplementary Tables 2 and 3. Commercial kits and enzymes were used according to the manufacturers' recommendations. Plasmid DNA and PCR amplicons were purified with the NucleoSpin™ plasmid EasyPure and NucleoSpin™ gel and PCR clean-up kits, respectively (Macherey-Nagel, Düren, Germany). PCR amplifications were performed using Phusion™ Hot Start II high-fidelity or Phusion™ U Hot Start DNA polymerases (Thermo Fisher Scientific Co., Waltham, MA, USA) if primers contained deoxyuracil residues. The OneTaq™ master mix (New England Biolabs, Ipswich, MA, USA) was used for colony PCRs. All FastDigest™ restriction enzymes and T4 DNA ligase (Thermo Fisher Scientific Co.) were used according to standard protocols. The USER enzyme (New England Biolabs) was used to perform USER-cloning procedures, and the Mix2Seq kit (Eurofins Genomics, Germany) was employed for routine Sanger sequencing.

To construct plasmid pBEC6, a *START* codon-less uracil DNA glycosylase inhibitor gene (*ugi*) from *Bacillus subtilis* bacteriophage AR9 (GenBank accession number YP_009283008.1) was codon-optimized for *Pseudomonas* species. This DNA fragment was then amplified with primers #4 and #5 (Supplementary Table 3), while vector pnCas9PA-BEC[41] was amplified with primers #6 and #7. The resulting fragments were joined by USER cloning. In order to insert a constitutively-expressed fluorescent marker upstream of the Cas9-handle, *msfGFP* was amplified from plasmid pBG42[95] with the primer pair #8 and #9, and ligated into the BsaI-digested pBEC6 backbone, giving rise to plasmid pMBEC6. Next, *cas6f* (*PA14_33300*) was amplified from the genome of *P. aeruginosa* PA14 with the primer pair #10 and #11, and cloned into the empty pSEVA434 vector[52] digested with the restriction enzymes EcoRI and BamHI. From the resulting pS434·Cas6f plasmid (Supplementary Table 1), a fragment containing $P_{trc}$-*cas6f*-$T_0$ was amplified with primers #12 and #13 and cloned with XbaI and XhoI into vector pBEC6 to yield the multiplex base editing plasmid pMBEC6. Finally, the Gm resistance marker in plasmid pMBEC6 was exchanged by resistance determinants for Km, Sm and Ap, generating plasmids pMBEC2, pMBEC4 and pMBEC8, respectively. A detailed protocol to construct the multiplex guide RNA is presented in Supplementary Methods. The CRISPY-web service[96] was employed to design the 20-nt-spacer sequence for each target gene. The GenBank sequences of *P. aeruginosa* PA14 (NC_008463.1), *P. putida* KT2440 (NC_008463.1), and *E. coli* (CP001509) were used to identify the spacers.

Plasmid pS2311·PCA was constructed by amplification of *quiC*, *aroQ* and *tktA* from the genome of *P. putida* with primer pairs #14 and #15; #16 and #17 and #18 and #19, respectively. Vector pSEVA2311 was amplified with primers #20 and #21. The resulting PCR products were used in a combined BsaI restriction/T4 ligation reaction similar to the protocol used for cloning gRNAs (File S2). Plasmid pFDH was constructed by PCR amplification of *fdh*ᵉ with the primer pair #22 and #23, and vector pSEVA621 with the primer pair #24 and #25. The fragments were fused by USER cloning as indicated above.

**Exploring editing efficiency through phenotypic analysis of *nicX*-inactivated mutants in *P. putida***. *Pseudomonas* strains were made electrocompetent by washing the biomass with a sucrose solution[97]. From 1 mL of the recovered cell culture, 100 µL aliquots were transferred to 10 mL of selective LB medium and grown for 24 h. When indicated, a subsequent passage of a 1:100 dilution was incubated for an additional 24 h. Thereafter, 10 mL of LB supplemented with sucrose at 10% (w/v) was inoculated with 100 µL of the final culture, and incubated overnight to cure the base-editing plasmid. Serial dilutions were then plated on non-selective LB plates. From these plates, colonies were randomly inspected for antibiotic sensitivity to verify plasmid curing and transferred to 200 µL of LB medium supplemented with 5 mM NA in 96-well plates. These plates was incubated for 24 h at 30 °C and 250 rpm, followed by incubation at 4 °C for 24 h.

Finally, the 96-well plates were visually inspected to identify positive colonies as a result of the pigment accumulated by *nicX*-negative clones[28].

**In silico prediction of editing efficiency in *Pseudomonas* genomes**. An accessible window of six bases, starting from position two of the protospacer, was implemented in a Python script to find putative editable protospacer sequences in the genomes of *P. putida* and *P. aeruginosa*[8,14] (Supplementary Data 1). The python script was executed in Spyder 3.3.6 with Python 3.7.4 and Biopython 1.76. Furthermore, all cytidines with a preceding guanidine were discarded as potential targets because of their low editing efficiency. Unique sequences were kept during protospacer screening and selection.

**Analytical procedures**. A 1 mL aliquot was withdrawn from cultures at the indicated time points. Samples were centrifuged at $13,000 \times g$ for 2 min and clarified supernatants were transferred to clean tubes. For PCA quantification, 10-µL samples were injected into a Supelco Discovery HS F5 column (150 mm × 4.6 mm × 3 µm; Sigma-Aldrich Co.) at 30 °C. Analytes in the samples were eluted with a buffer composed of 95% (v/v) of 10 mM ammonium formate and 5% (v/v) acetonitrile at a flow of 0.7 mL min⁻¹. The acetonitrile concentration was kept constant for 0.5 min and then linearly increased to 60% over 4.5 min. Subsequently, the acetonitrile concentration was increased to 90% over 0.5 min and washed with 90% for 2 min. After reducing the concentration to 5% over 0.1 min, the column was re-equilibrated with 5% (v/v) acetonitrile for 2.4 min. PCA was detected at 260 nm, and its chemical identity was confirmed with authentic standards. For sugar analysis, 20 µL of the supernatant was injected onto an Aminex HPX-87H column (300 mm × 7.8 mm × 9 µm, BioRad). The sample was eluted over 30 min at 30 °C with 5 mM $H_2SO_4$ at a flow of 0.6 mL min⁻¹. Refraction index was used for glucose detection, while absorbance at 205 nm was used to quantify gluconate and 2-ketogluconate.

**Whole genome sequencing and polymorphism analysis**. Genomic DNA was purified using the PureLink™ Genomic DNA purification kit (Invitrogen, Waltham, MA, USA) from 2 mL of overnight LB cultures of isolated clones as described previously[98–101]. DNA was randomly sheared into short fragments of ca. 350 bp using ultrasonic interruption. Short and large DNA fragments were removed using magnetic bead size selection and subsequently verified by capillary gel electrophoresis. The obtained DNA fragments were subjected to library construction using the NEBNext™ DNA Library Prep Kit (New England Biolabs), following the supplier's specifications. Libraries quality control was performed with a Qubit 2.0 fluorometer and an Agilent™ 2100 BioAnalyzer. Subsequent sequencing was performed using the Illumina NovaSeq™ 6000 PE150 platform. The original sequencing data acquired by high-throughput sequencing platforms recorded in image files were transformed to sequence reads by base calling with the Illumina's CASAVA software. Sequences and associated quality information were stored in a FASTQ file. For quality-control purposes, paired reads were discarded when: (i) either read contained adapter contamination, (ii) uncertain nucleotides (N) constituted >10% of either read, or (iii) low quality nucleotides (base quality ≤ 5) constituted >50% of either read. The effective sequencing data were aligned with the *P. putida* KT2440 reference sequence[21] (GenBank, NC_002947), after error correction, trimming, and normalization through Geneious (Biomatters, New Zealand), and the mapping rate and coverage were counted according to the alignment results. Single nucleotide polymorphisms (SNPs) and InDels were detected in Geneious. Variant calling of single nucleotide polymorphisms (SNPs) and small insertions and deletions (indels) was performed with a variant frequency threshold of 0.4 on all samples. Additionally, a variant calling with a threshold of 0.1 was performed on a negative control sample (i.e., not subjected to base editing) to identify existing nucleotide polymorphisms in the parental strain. The construction of libraries, sequencing, and subsequent data quality control was performed by Novogene Co. Ltd. (Cambridge, United Kingdom).

**Data and statistical analysis**. All the experiments reported were independently repeated at least twice (as indicated in the corresponding figure or table legend), and the mean value of the corresponding parameter ± standard deviation is presented. In some cases, the level of significance of the differences when comparing results were evaluated by means of the Student's $t$ test with $\alpha = 0.01$ or $\alpha = 0.05$ as indicated in the figure legends.

**Reporting summary**. Further information on research design is available in the Nature Research Reporting Summary linked to this article.

## Data availability
Data supporting the findings of this work are available within the paper and its Supplementary Information files. The genome sequences used for *P. aeruginosa* PA14 (NC_008463.1), *P. putida* KT2440 (NC_002947), and *E. coli* (CP001509) are accessible in GenBank. The plasmids created in this work can be requested through Addgene. A reporting summary for this article is available as a Supplementary Information file. The datasets generated and analyzed during the current study are provided in the Source data.

The sequence from whole genome sequencing of the genome-edited strains is available at NCBI in the BioProject platform with accession number PRJNA836759.

## Code availability
All scripts are provided with this paper in Supplementary Data 1.

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

## Acknowledgements

The financial support from The Novo Nordisk Foundation through grants NNF20CC0035580, *LiFe* (NNF18OC0034818) and *TARGET* (NNF21OC0067996), the Danish Council for Independent Research (*SWEET*, DFF-Research Project 8021-00039B), and the European Union's Horizon 2020 Research and Innovation Program under grant agreement No. 814418 (*SinFonia*) to P.I.N. is gratefully acknowledged. E.K. was supported by the Novo Nordisk Foundation (grant NNF17CC0026768) as part of the Copenhagen Bioscience Ph.D. Program. A.M.S. received funding from Agencia Nacional de Promoción Científica y Tecnológica (ANPCyT, PICT-2016-1545 and PICT-2019-1590). R.A.M. was supported by a postdoctoral fellowship from ANPCyT and a Research and Training Grant from the Federation of European Microbiological Societies (FEMS). The responsibility of this article lies with the authors, and the funding sources are not responsible for any use that may be made of the information contained therein.

## Author contributions

D.C.V., R.A.M., and P.I.N. conceived the project and planned the experiments. D.C.V., R.A.M., and E.K. performed the experiments and provided all the materials and protocols described in the article, including the testing of the base editor in different species. All the authors analyzed the data and participated in the discussions that led to the conclusions of the study. D.C.V. and P.I.N. wrote the manuscript, with contributions of all other authors.

## Competing interests

The authors declare no competing interests.
