## [Peer Review File · Nature Communications]

Reviewers' Comments:

Reviewer #1:

Remarks to the Author:

General comments;

CRISPR-based DNA base editing tools enable single base conversion without generating DNA double strand breaks (DSBs), which is useful for genome editing in bacteria that do not have a non-homologous end joining pathway. In this manuscript, the authors optimized a multiplex cytosine base editing system by using Cas6 for cleaving pre-gRNA transcripts. Although the authors developed a practical base editing system for Gram-negative bacteria, I do not think that it is much advanced compared to pre-existing toolkits, thus it might be suitable to more specific journal, rather than this journal, Nature Communications.

Specific Comments:

1) First of all, it is very confusing on terminologies. Does CBE indicate a cytosine base editor or CRISPR base editor? Generally, DNA Base Editors (BEs) comprise a cytosine base editor (CBE) and an adenine base editor (ABE). And CBE consists of nCas9 and a cytidine deaminase with or without a uracil glycosylase inhibitor (UGI), whereas ABE consists of nCas9 and an adenosine deaminase. It is necessary to revise the terms throughout the manuscript.

2) Previously, several groups have already showed a CBE-mediated gene disruption by creating premature stop codon within ORFs in many species including human and bacteria. What is the uniqueness of this study compared to others? Is it a multiple base editing using Cas6? But this concept has also been reported in several papers. Although I fully understand that this manuscript is highly practical in terms of detailed protocols such as the Golden Gate assembly of multiple gRNAs and the one-step engineering tool, it is necessary to emphasize the uniqueness or strength in the discussion section to reach the criteria of this journal.

3) In Figure 2, the authors claimed that the CBE showed no editing effect in 'GC motif', but I think that the targets in Figure 2D was biased and abnormal. The targets are too GC-rich. I think that the authors should test CBE activity in arbitrary target sites with 40~60% of GC contents such as '-NNNGCNNN-', instead of 'GCGCGCGC---'. In ref #8, the Liu group showed that CBE was less effective in GC motif but worked anyway.

4) In page 6, the authors argued that the presence of UGI raises the occurrence of off-target events but, to my knowledge, the cytidine deaminase of CBE induces genome-wide off-target effects even in the absence of gRNAs [Science, DOI: 10.1126/science.aav9973]. It should be modified.

4-1) As mentioned above, the authors should investigate the CBE-mediated genome-wide off-target effects using whole genome sequencing.

4-2) The UGI rather affect the base conversion purity. For example, in the presence of UGI, cytosines are usually converted to thymines, whereas cytosines are converted to guanines or adenines in the absence of UGI. It is necessary to reveal the base editing purities with or without UGI in Figure 2.

Reviewer #2:

Remarks to the Author:

The paper by Volke et al. describes the development of a (multiplexed) CRISPR-based, base-editing tool for Gram-negative bacteria. The tool is based on the type II-A CRISPR-Cas effector nuclease Cas9 from *Streptococcus pyogenes* (SpyCas9), which is an RNA-guided, DNA-targeting nuclease. The authors fused the APOBEC1 cytidine deaminase from *Rattus norvegicus* to the N-terminus of a nickase mutant of SpyCas9 (nickase SpyCas9; nSpyCas9D10A) to enable targeted cytidine-to-adenine deamination for the creation of a premature STOP codon in selected ORF(s). Moreover, they fused the uracil glycosylase inhibitor (UGI) from the *Bacillus bacteriophage* AR9 to the C-terminus of nSpyCas9D10A to increase the base-editing efficiency and broaden the editing

window.

CRISPR-guided, base-editing tools ('BEs') are widely applied nowadays in both prokaryotes and eukaryotes mainly for the generation of premature STOP codons in genes or introduce non-synonymous mutations to disrupt the function of the encoded protein. BEs are grouped into cytidine base-editors ('CBEs') and adenosine base-editors ('ABEs'). CBEs contain a cytidine deaminase variant that converts cytidine-to-uracil, which is subsequently converted into thymine during DNA replication/repair. ABEs contain an adenosine deaminase variant that converts adenine-to-inosine, which is read as guanine on DNA level during translation and DNA replication.

In this study, Volke et al. demonstrated in silico that application of a CBE tool for the creation of early, premature STOP codons is possible for 92% of genes in *Pseudomonas putida* KT2440 and 75% of genes in *Pseudomonas aeruginosa* PA14. They edited single genes of *P. putida* with high efficiencies, demonstrated the nucleotide context preference for editing by APOBEC and showed that UGI increased both the editing efficiency and the editing window. Next, they successfully developed and tested a multiplex editing setup to edit multiple genes of *P. putida*, *P. aeruginosa* and *Escherichia coli*. This was achieved by combining their tool with the type I-F CRISPR-associated endoribonuclease Cas6 (Cas6f) which processes a long RNA transcript (containing multiple gRNAs separated by Cas6 cleavage sites) into individual mature gRNAs (each targeting a different gene). As a proof of concept to '(de)construct' phenotypes with their multiplex editing tool, *P. putida* was optimised for the production of aromatic compounds, and a synthetic 'NADPH-auxotrophic' strain was created.

In terms of novelty for the tool development part, the study is reasonable advancement over prior (non-multiplexed) base editing work in *Pseudomonas* (e.g. Chen et al., *iScience* 2018) and reconfirms several findings (enhancement of base editing with UGI, nucleotide context preference for editing by APOBEC) and design strategies (the use of Cas6 to generate multiple gRNAs for multiplexing, e.g. Tsai et al., *Nat Biotechnol.* 2014) made earlier. In this reviewer's opinion, the strongest aspects of this study are 1) the authors' efforts to streamline the plasmid assembly (for multiplex gene editing) and providing details descriptions/protocols on how to use it and 2) their elaborate (metabolic) engineering work presented in Figures 5 and 6 as a proof of concept for the strength of their CBE tool. The experiments are well conducted, the conclusions are in good agreement with the data presented and the manuscript is overall well written (exceptions listed below).

Major comments

The authors should (more elaborately) acknowledge prior published work on 1) base editing design and execution in *Pseudomonas* (Chen et al., *iScience* 2018), 2) the conclusions on the effect on editing efficiency and nucleotide preference for UGI (Komor et al., *Nature* 2016) and 3) the use of Cas6 to generate mature gRNAs (Tsai et al., *Nat Biotechnol.* 2014). For the latter, the authors should also elaborate whether they make use of the same design principles as the Tsai et al study, as this is currently unclear. Also, why are there 2 stem-loops in the pre-crRNA?

The authors consider editing successful "if at least one cytidine was altered to thymidine in the protospacer sequence" (P8. L4-6). Since the readout was done with Sanger sequencing, can the authors elaborate on whether the conversion in all of their experiments was 100% (single peak) or whether mixed-peaks in the chromatograms were observed? At the very least, the authors should include the chromatograms on which these conclusions were based. They should also specify how many individual clones were sequenced for the section described on P8, L3-4. Lastly, did the authors have an indication of potential off-targeting events in their base editing setup (e.g. low CFUs when compared to a non-targeting spacer)?

In contrast to the rest of the manuscript, the conclusion section is underwhelming and lacks depth and focus. In its current state, two-thirds of the text is spent on a mathematical calculation on the theoretical probability of mutation reversion, which is (in this reviewer's opinion) too much given the wealth of interesting data in the manuscript, that could have been discussed here instead.

The results presented in Figure 3 carry the title "Testing multiplex base editing across bacterial species". However, only multiplexing (efficiencies) were tested for E.coli BL21 (and P. aeruginosa PA14), but not for P. putida KT2440. The authors should determine and discuss the editing efficiencies of the other 4 targets (benA, gclR, glpR, and nfxB).

The authors use the abbreviation CBE for "CRISPR base editors", which might be confusing to different readers, as it is widely used in the literature of the base-editing field to abbreviate 'cytidine base-editors'. A suggestion would be to use the commonly accepted abbreviation 'BEs' to describe CRISPR base-editors.

Minor comments

On several occasions, the authors refer to P. putida as a "non-traditional bacterial platform". It is unclear why the authors make this statement as the physiology, metabolism, and genetics are well-studied, and several genetic engineering tools have been developed.

I would recommend to have another careful look at the grammar and clearness of certain words/phrases, such as "on this background", "the practical editing region afforded by pBEC plasmids", "depiecing" (is this different from "deconstructing"?), "In all case, ...", "Upon editing, the target loci in were amplified", "λ-Red recombineering, commonplace for E. coli engineering.", "from mammalian cells over yeast to bacterial species", etc

P4, L19: "Starting by a thorough characterization of the exact editing window and stabilizing RNA motifs together with the influence of incubation times, the editing performance of a synthetic CBE was further enhanced by engineering a uracil glycosylase inhibitor (ugi)."
Please elaborate what is meant with "stabilizing RNA motifs"

P6, L15: "Introducing UGI in CBEs increased editing efficiency in mammalian cells and E. coli, but could potentially raise the occurrence of off-target events."
In the referred article, it mentions that the off-target (e.g unwanted indels and substitutions) were higher in the delta-UGI strain. Perhaps the authors meant the opposite here?

P7, L9: "this endoribonuclease can be utilized to process a single polycistronic mRNA molecule into several individual, mature gRNAs"
Please revise (also at other instances throughout the manuscript). A pre-crRNA is neither polycistronic (it does not encode for multiple polypeptides), nor is it an mRNA.

P7, L27: "The target 20-nt protospacers were chosen such that a cytidine residue alternated with another base in the sequence, i.e., 5'-NCN CNC NCN C-3' (Fig. 2D and Table S2)."
The authors should describe more clearly what was tested here. The current description gives the impression that only cytidines at positions 2,4,6 and 8 were tested.

P8, L11: "We verified editing with a guanidine preceding the target base, but this was not the case for guanine-rich protospacers—which indicates that the editing efficiency is affected not only by the preceding base(s) but also by the subsequent residues in the 3'-end of the target."
Please revise, as this is somewhat confusing (what is "this" referring to?)

P8, L25-L26: "Based on these observations, the practical editing region afforded by pBEC plasmids was assigned to 8 nt (i.e., positions 2 to 9 of the protospacer) in the PAM-distal sequence (Fig. 2F)."
It is not clear which was the editing window without UGI compared to the presence of UGI.

P10 and P11: the authors attempt to rationalize the increase in editing efficiencies upon adding a Cas6 recognition site to the most distal spacer. Did the authors also consider whether protective 3' end groups (3' phosphate or cyclic phosphate) which are typically for these endoribonucleases (e.g. Carte et al., Mol Microbiol. 2015) might be responsible for this?

P18, L10-L12: "Importantly, our CBE technology enables target, multiplex modifications in non-

traditional microbial hosts, e.g., *Pseudomonas* species, where tools commonly applied to model bacteria, e.g., λ -Red recombineering, display very low efficiencies.”
Reference(s) is/are missing.

Legend Figure 2: the title of the legend needs to reflect what is being depicted, namely editing in *P. putida* KT2440, not “gram-negative bacteria”

Legend panel 2D: “All bases are represented in all positions within the editing window of the base editor.”

Please describe more clearly what is meant here. Also, what is indicated in the grayed area?

Figure 4 (and other instances). The figure and text throughout the manuscript gives the impression that the spacer is not part of the gRNA, which is not the case. Please correct this.

Figure 5A: change “edition” to “editing”

Response to the Reviewers' comments and how they have been addressed in the revised version of NCOMMS-21-47278 (in blue)

Dear Referees,

Many thanks for taking the time to evaluate our manuscript and for your constructive reviews of our study. The overall positive comments and the useful suggestions encouraged us to put substantial additional work into preparing a resubmission. We have modified the manuscript and supplementary files in a way that has addressed the points you have raised during the first round of revision. We have added extra experimental evidence, providing new and revised figures as well as additional explanations in the text, which we believe have enhanced the clarity of the message and improved our manuscript. All specific comments and our replies to them are listed below (the comments have been numbered to facilitate perusal). The corresponding modifications in the accompanying article are highlighted in yellow for easy tracking. We hope you are as satisfied with the revised form of this manuscript as we are!

Referee # 1

“CRISPR-based DNA base editing tools enable single base conversion without generating DNA double strand breaks (DSBs), which is useful for genome editing in bacteria that do not have a non-homologous end joining pathway. In this manuscript, the authors optimized a multiplex cytosine base editing system by using Cas6 for cleaving pre-gRNA transcripts. Although the authors developed a practical base editing system for Gram-negative bacteria, I do not think that it is much advanced compared to pre-existing toolkits, thus it might be suitable to more specific journal, rather than this journal, Nature Communications”

We thank the Reviewer for the candid evaluation of our manuscript. We agree in that other DNA base editors have been published recently, although we are convinced that our current work brings the application of these tools to a level that has not been attempted hitherto. Besides the in-depth characterization of the base editing system and the combination of multiplex base editing with very high editing efficiency, a key message of our article is the development of a robust and streamlined process from *in silico* design of spacer, over cloning to construction and verification of the desired (DNA-edited) microbial strains. Our protocol allows for the application of the technology in non-traditional bacterial species for *complex* genotype engineering, which were thus far not possible with the existing genome engineering protocols. This is illustrated both by (i) constructing a PCA-producing *P. putida* strain in a single engineering step, and (ii) by disentangling (deconstructing) the intricate redox metabolism of this species. In both cases, the extant toolset for genome manipulations would be simply no sufficient for these purposes. Furthermore, we show that incorporating Cas6 in the base editor tool significantly increases efficiency. Taken these aspects together, we believe that these efforts will pave the way for a new wave of complex metabolic engineering projects in several species (including *Pseudomonas*, and beyond). Therefore, these results will be highly relevant for researchers in the fields of synthetic biology, metabolic engineering, and microbiology—and we are convinced that *Nature Communications* is an appropriate journal to share our results with such a diverse scientific community.

- 1.1 *“First of all, the article is very confusing on terminologies. Does CBE indicate a cytosine base editor or CRISPR base editor? Generally, DNA Base Editors (BEs) comprise a cytosine base editor (CBE) and an adenine base editor (ABE). CBE consists of nCas9 and a cytidine deaminase with or without a uracil glycosylase inhibitor (UGI), whereas ABE consists of nCas9 and an adenosine deaminase. It is necessary to revise the terms throughout the manuscript.”*

Many thanks for pointing out this matter. We abide by the common terminology in the field and, according to the Reviewer's suggestion, we now use *DNA Base Editor* (BE), *cytosine base editor* (CBE) and *adenine base editor* (ABE) to refer to the different editors. We have revised the nomenclature throughout the manuscript accordingly.

1.2 “Previously, several groups have showed a CBE-mediated gene disruption by creating premature stop codons within ORFs in many species including human and bacteria. What is the uniqueness of this study compared to others? Is it a multiple base editing using Cas6? But this concept has also been reported in several papers. Although I fully understand that this manuscript is highly practical in terms of detailed protocols such as the Golden Gate assembly of multiple gRNAs and the one-step engineering tool, it is necessary to emphasize the uniqueness or strength in the discussion section to reach the criteria of this journal.”

We thank for this critical remark. We agree that emphasis should be put on practical applications of this technology and its advantages over the existing protocols—and we have re-worked the discussion section to highlight these points. As indicated in the first general comment, we believe that the in-depth calibration of the tool in non-traditional hosts is key to unleash its full potential for gene edition in bacteria (an aspect that has been only partially addressed in other articles). Most importantly, the two complex engineering programmes described in our article (i.e., production of aromatic compounds and depiecing the redox metabolism of *P. putida*) would have not been possible without this CBE toolset, and thus we think that the broad synthetic biology, metabolic engineering, and microbiology research communities would benefit from access to the technology.

1.3 “In Figure 2, the authors claimed that the CBE showed no editing effect in ‘GC motif’, but I think that the targets in Figure 2D are biased. The targets are too GC-rich. I think that the authors should test CBE activity in arbitrary target sites with 40~60% of GC contents such as ‘-NNNGCNNN-’, instead of ‘GCGCGCGC---’. In ref. # 8, the Liu group showed that CBE was less effective in GC motif but worked anyway.”

Agreed. The guanine content in the protospacer is high due to the criteria adopted for their design; we pointed this argument out in P8L6-15. However, we aimed at keeping the protospacer comparable across conditions for precise calibration of the toolset, and we also investigated the impact of non-adjacent bases as this aspect has not been explored before. This led to the finding that there is a negative effect (probably due to steric hindrance) of guanine and adenine-rich spacer regions. Also, please note that these GC-rich targets are typical of organisms like *P. putida* (and other soil bacteria, such as *Streptomyces*), and therefore their investigation is relevant when using the tool in hosts beyond *E. coli* and similar ‘model’ organisms. Additionally, and specifically requested by the Reviewer, we added new results of a less-G-rich spacer in the Supplementary material (Fig. S2). The results of this base editing experiment (editing the *mutS* gene of *P. putida*) indicate that –NNNGCNNN– are equally accessible (discussed in P8).

1.4 “In P6, the authors argued that the presence of UGI raises the occurrence of off-target events but, to my knowledge, the cytidine deaminase of CBE induces genome-wide off-target effects even in the absence of gRNAs [Science, DOI: 10.1126/science.aav9973]. This should be modified.”

1.4.1 “As mentioned above, the authors should investigate the CBE-mediated genome-wide off-target effects using whole genome sequencing.”

We agree with the Reviewer that there is a probability for genome-wide off-target effects, as the CBE will also access single-stranded DNA unrelated to the unwinding of the DNA by Cas9. DNA replication is one

such process. However, the uracil DNA glycosylase inhibitor may also raise the overall mutation rate in the cell, as it blocks DNA repair in a Cas9-unrelated mechanism. To investigate the potential introduction of genome-wide off-target mutations by the CBE, we conducted whole-genome sequencing (WGS) of several engineered strains in our study, including the PCA-producing *P. putida* strain and the redox-deficient *P. putida* strain. The results indicate that the CBE alone leads to a roughly doubling of the basal mutation frequency. The additional expression of UGI, on the other hand, leads to three to four fold increase in the mutation rate. These results are accordance with the publication cited by the Reviewer [DOI: 10.1126/science.aav9973]. The CBE in this publication carries UGI and is shown to induce off-target effects, while the ABE editor does not carry a UGI and has only a marginal off-target mutation rate—and our new data discriminates these effects by testing a CBE with and without UGI. The expression of UGI alone is known to raise the mutation frequency in bacteria [DOI:10.1074/jbc.M302121200], in line with our observations. The data from WGS is now included in the manuscript and supplementary data, and the results section of the article has been updated to include these observations together with the relevant references as requested (please see P15, P20 and File S5).

1.4.2 *“The UGI rather affect the base conversion purity. For example, in the presence of UGI, cytosines are usually converted to thymines, whereas cytosines are converted to guanines or adenines in the absence of UGI. It is necessary to reveal the base editing purities with or without UGI in Figure 2.”*

Thanks for this valuable remark. We only observed C→T transitions, no matter if UGI was expressed or not. These passages in the manuscript have been revised accordingly to reflect our experimental observations on base conversion purity (P9L14-18).

Referee # 2

*“The paper by Volke et al. describes the development of a (multiplexed) CRISPR-based, base-editing tool for Gram-negative bacteria. The tool is based on the type II-A CRISPR-Cas effector nuclease Cas9 from *Streptococcus pyogenes* (SpyCas9), which is an RNA-guided, DNA-targeting nuclease. The authors fused the APOBEC1 cytidine deaminase from *Rattus norvegicus* to the N-terminus of a nickase mutant of SpyCas9 (nickase SpyCas9; nSpyCas9D10A) to enable targeted cytidine-to-adenine deamination for the creation of a premature STOP codon in selected ORF(s). Moreover, they fused the uracil glycosylase inhibitor (UGI) from the *Bacillus bacteriophage* AR9 to the C-terminus of nSpyCas9D10A to increase the base-editing efficiency and broaden the editing window.*

CRISPR-guided, base-editing tools (‘BEs’) are widely applied nowadays in both prokaryotes and eukaryotes mainly for the generation of premature STOP codons in genes or introduce non-synonymous mutations to disrupt the function of the encoded protein. BEs are grouped into cytidine base-editors (‘CBEs’) and adenosine base-editors (‘ABEs’). CBEs contain a cytidine deaminase variant that converts cytidine-to-uracil, which is subsequently converted into thymine during DNA replication/repair. ABEs contain an adenosine deaminase variant that converts adenine-to-inosine, which is read as guanine on DNA level during translation and DNA replication.

*In this study, Volke et al. demonstrated in silico that application of a CBE tool for the creation of early, premature STOP codons is possible for 92% of genes in *Pseudomonas putida* KT2440 and 75% of genes in *Pseudomonas aeruginosa* PA14. They edited single genes of *P. putida* with high efficiencies, demonstrated the nucleotide context preference for editing by APOBEC and showed that UGI increased both the editing efficiency and the editing window. Next, they successfully*

developed and tested a multiplex editing setup to edit multiple genes of *P. putida*, *P. aeruginosa* and *Escherichia coli*. This was achieved by combining their tool with the type I-F CRISPR-associated endoribonuclease Cas6 (Cas6f) which processes a long RNA transcript (containing multiple gRNAs separated by Cas6 cleavage sites) into individual mature gRNAs (each targeting a different gene). As a proof of concept to '(de)construct' phenotypes with their multiplex editing tool, *P. putida* was optimized for the production of aromatic compounds, and a synthetic 'NADPH-auxotrophic' strain was created.

In terms of novelty for the tool development part, the study is reasonable advancement over prior (non-multiplexed) base editing work in *Pseudomonas* (e.g. Chen et al., *iScience* 2018) and reconfirms several findings (enhancement of base editing with UGI, nucleotide context preference for editing by APOBEC) and design strategies (the use of Cas6 to generate multiple gRNAs for multiplexing, e.g. Tsai et al., *Nat Biotechnol.* 2014) made earlier. In this reviewer's opinion, the strongest aspects of this study are 1) the authors' efforts to streamline the plasmid assembly (for multiplex gene editing) and providing details descriptions/protocols on how to use it and 2) their elaborate (metabolic) engineering work presented in Figures 5 and 6 as a proof of concept for the strength of their CBE tool. The experiments are well conducted, the conclusions are in good agreement with the data presented and the manuscript is overall well written (exceptions listed below)."

We thank the Reviewer for the critical evaluation of our work and his/her valuable input, which has been addressed as indicated in the specific points below.

2.1 "The authors should (more elaborately) acknowledge prior published work on 1) base editing design and execution in *Pseudomonas* (Chen et al., *iScience* 2018), 2) the conclusions on the effect on editing efficiency and nucleotide preference for UGI (Komor et al., *Nature* 2016) and 3) the use of Cas6 to generate mature gRNAs (Tsai et al., *Nat. Biotechnol.* 2014). For the latter, the authors should also elaborate whether they make use of the same design principles as the Tsai et al. study, as this is currently unclear."

Thank you for drawing our attention to the relevant literature. These citations were included in our original submission, but now we have emphasized prior work on base-editing further. Furthermore, the design of the toolbox and similarities to previous studies have been added and discussed as requested (P4 and P7).

2.1.1 "Also, why are there 2 stem-loops in the pre-crRNA?"

Thanks. In this study, a synthetic fusion of CRISPR RNA (crRNA) and the *trans*-activating CRISPR RNA (tracrRNA) is used. This aspect is now explicitly indicated in the manuscript (P6). In chimeric RNAs, one stem-loop mimics the crRNA-tracrRNA binding (Cas9 handle), whereas the second stem-loop is identical to the stem-loop in the natural tracrRNA (truncated *S. pyogenes* terminator).

2.2 "The authors consider editing successful "if at least one cytidine was altered to thymidine in the protospacer sequence" (P8, L4-6). Since the readout was done with Sanger sequencing, can the authors elaborate on whether the conversion in all of their experiments was 100% (single peak) or whether mixed-peaks in the chromatograms were observed?"

We are grateful for the detailed feedback. Cultures were grown after introducing the base-editing plasmid for several generations (24 h) in order to facilitate editing and for subsequent curing of the plasmid. This procedure fostered genotypic segregation and single peaks were almost exclusively observed in the

chromatograms. In the rare cases where double peaks were observed, the ratio between the peaks was taken into account to calculate the abundance of the resulting genotype. This information is now explicitly detailed in the revised manuscript (P8).

2.2.1 *“At the very least, the authors should include the chromatograms on which these conclusions were based. They should also specify how many individual clones were sequenced for the section described on P8, L3-4.”*

The chromatograms of these experiments are now included in the Supplementary document. The number of sequenced clones is specified in the caption to Figure 2 (“Mean values of two independent biological experiments are presented; dots represent data per experiment with at least eight colonies analyzed per replicate”). Furthermore, chromatograms for one protospacer and the full set of sequencing data for one of the base-editing experiments are now presented in Figure S1.

2.2.2 *“Lastly, did the authors have an indication of potential off-targeting events in their base editing setup (e.g. low CFUs when compared to a non-targeting spacer)?”*

That is an excellent question. We did not observe any indication of toxicity of the construct (e.g. the growth rate of the strains transformed with the base-editing plasmid was not different as compared to a control strain carrying the empty pSEVA631 vector). We now include whole-genome sequencing to identify potential off-targeting effects (please see the response to comment 4 by Reviewer # 1). The mutation rates are increased in cells carrying the CBE alone, and further increased to two to three-fold in the case that the base editor includes UGI. These results (including those shown in File S5) are discussed in P15 and P20.

2.3 *“In contrast to the rest of the manuscript, the conclusion section is underwhelming and lacks depth and focus. In its current state, two-thirds of the text is spent on a mathematical calculation on the theoretical probability of mutation reversion, which is (in this Reviewer’s opinion) too much given the wealth of interesting data in the manuscript, that could have been discussed here instead.”*

We thank the Reviewer for the suggestion. Fully agreed. We revised the Discussion section to focus on the advances in the toolbox and the construction of complex phenotypes enabled by this technology—we hope that the current version of the Discussion has the same standard and quality as the rest of the manuscript.

2.4 *“The results presented in Figure 3 carry the title “Testing multiplex base editing across bacterial species”. However, only multiplexing (efficiencies) were tested for E. coli BL21 (and P. aeruginosa PA14), but not for P. putida KT2440. The authors should determine and discuss the editing efficiencies of the other 4 targets (benA, gclR, glpR, and nfxB).”*

Thank you for pointing this mishap. We now include the overall editing efficiency of the final 5-spacer construct in *P. putida* KT2440 in the figures and the text (P12) as requested by the Reviewer.

2.5 *“The authors use the abbreviation CBE for “CRISPR base editors”, which might be confusing to different readers, as it is widely used in the literature of the base-editing field to abbreviate ‘cytidine base-editors’. A suggestion would be to use the commonly accepted abbreviation ‘BEs’ to describe CRISPR base-editors”*

Thank you for this remark. The nomenclature was changed throughout the manuscript for the sake of consistency and clarity (see also the response to comment 1.1 by Referee # 1).

- 2.6 *“On several occasions, the authors refer to P. putida as a “non-traditional bacterial platform”. It is unclear why the authors make this statement as the physiology, metabolism, and genetics are well-studied, and several genetic engineering tools have been developed.”*

We agree with the reviewer that *P. putida* is not as exotic a bacterium as it used to be a couple of years back. Nevertheless, the toolbox and the fundamental knowledge of this host are still lagging behind in comparison to ‘classical’ model organisms like *E. coli* and *S. cerevisiae*. We have modified the text to make this distinction clearer in P4 and the Conclusion.

- 2.7 *“I would recommend to have another careful look at the grammar and clearness of certain words/phrases, such as “on this background”, “the practical editing region afforded by pBEC plasmids”, “depiecing” (is this different from “deconstructing”?), “In all case, ...”, “Upon editing, the target loci in were amplified”, “λ-Red recombineering, commonplace for E. coli engineering.”, “from mammalian cells over yeast to bacterial species”, etc.”*

Thanks for point this issue. The text has been combed out of grammatically-unsound and unclear sentences (probably arising from the fact that none of the authors is a native English speaker) to the best of our abilities. We had the manuscript read by colleagues proficient in the language and we hope that the current version of the article is clearer and sharper in its message.

- 2.8 *“P4, L19: “Starting by a thorough characterization of the exact editing window and stabilizing RNA motifs together with the influence of incubation times, the editing performance of a synthetic CBE was further enhanced by engineering a uracil glycosylase inhibitor (ugi).”Please elaborate what is meant with “stabilizing RNA motifs””*

Thanks! These sentences were rephrased and an explanation has been added to explain what is meant in this part of the manuscript (P4).

- 2.9 *“P6, L15: “Introducing UGI in CBEs increased editing efficiency in mammalian cells and E. coli, but could potentially raise the occurrence of off-target events.” In the referred article, it mentions that the off-target (e.g. unwanted indels and substitutions) were higher in the delta-UGI strain. Perhaps the authors meant the opposite here?”*

We referred to off-target effects as mutations outside the protospacer region (indicated as “off-target events” in the manuscript). The off-target mutations are discussed in questions 1.4 and 2.2.2, and the corresponding information is now added to the article. We did not observe other mutations than C→T in the protospacer region, with or without UGI; this is now explicitly indicated in the text in P9L14.

- 2.10 *“P7, L9: “this endoribonuclease can be utilized to process a single polycistronic mRNA molecule into several individual, mature gRNAs” Please revise (also at other instances throughout the manuscript). A pre-crRNA is neither polycistronic (it does not encode for multiple polypeptides), nor is it an mRNA.”*

This passages has been edited in the revised article to correct this mishap.

- 2.11 *“P7, L27: “The target 20-nt protospacers were chosen such that a cytidine residue alternated with another base in the sequence, i.e., 5'-NCN CNC NCN C-3' (Fig. 2D and Table S2).” The authors should describe more clearly what was tested here. The current description gives the impression that only cytidines at positions 2, 4 ,6 and 8 were tested.”*

Agreed, the original sentence was not completely clear, and an explanation was added to the manuscript for clarity (P8L3).

- 2.12 *“P8, L11: “We verified editing with a guanidine preceding the target base, but this was not the case for guanine-rich protospacers—which indicates that the editing efficiency is affected not only by the preceding base(s) but also by the subsequent residues in the 3'-end of the target.” Please revise, as this is somewhat confusing (what is “this” referring to?)”*

This paragraph has been revised as suggested (P8L24).

- 2.13 *“P8, L25-L26: “Based on these observations, the practical editing region afforded by pBEC plasmids was assigned to 8 nt (i.e., positions 2 to 9 of the protospacer) in the PAM-distal sequence (Fig. 2F).” It is not clear which was the editing window without UGI compared to the presence of UGI.*

We added the missing information to the manuscript (see also comments above on the presence of UGI, P9).

- 2.14 *“P10 and P11: the authors attempt to rationalize the increase in editing efficiencies upon adding a Cas6 recognition site to the most distal spacer. Did the authors also consider whether protective 3' end groups (3' phosphate or cyclic phosphate) which are typically for these endoribonucleases (e.g. Carte et al., Mol. Microbiol. 2015) might be responsible for this?”*

Many thanks for the suggestion; the mechanism indicated by the Reviewer is also a possible explanation for the observed patterns in editing efficiency. However, revealing the exact nature of the mechanism that leads to higher efficiency in the presence of Cas6 is experimentally difficult to elucidate. This is due to the fact that the 3'-modification and binding of Cas6 to the RNA happen simultaneously. We have included this hypothesis as a potential explanation in the manuscript in P11.

- 2.15 *“P18, L10-L12: “Importantly, our CBE technology enables target, multiplex modifications in non-traditional microbial hosts, e.g., Pseudomonas species, where tools commonly applied to model bacteria, e.g., λ -Red recombineering, display very low efficiencies.” Reference(s) is/are missing.”*

References for this claim have been added in P4L10.

- 2.16 *“Legend Figure 2: the title of the legend needs to reflect what is being depicted, namely editing in P. putida KT2440, not “gram-negative bacteria”. Legend panel 2D: “All bases are represented in all positions within the editing window of the base editor.” Please describe more clearly what is meant here. Also, what is indicated in the grayed area?”*

The legend for Figure 2 was modified as suggested. The grayed area serves as a guide for the reader to easily identify positions within the protospacer.

- 2.17 *Figure 4 (and other instances). The figure and text throughout the manuscript gives the impression that the spacer is not part of the gRNA, which is not the case. Please correct this.*

Thank you. We changed the phrasing and the figures to clarify that the spacer is part of the gRNA.

2.18 *Figure 5A: change “edition” to “editing”*

The word has been changed. Additionally, the same wording was changed in P18 for consistency.

Reviewers' Comments:

Reviewer #1:

Remarks to the Author:

The authors have mostly answered the issues I raised in the earlier review. However, I still think that it might not reach the criteria of this journal, Nat Commun, because of the low technical novelty.

Reviewer #2:

Remarks to the Author:

The authors have substantially improved the quality of their manuscript by incorporating the suggestions made by the reviewers, adding supplementary data that was missing from the initial submission of the manuscript and conducted extra experiments to demonstrate the genome-wide off-targeting effects of their base-editor.

One last concern is that I could not find File S5 on the submission portal of the revised manuscript, but perhaps this is just a technical issue (filesize too large)?

**Response to the Reviewers' comments and how they have been addressed
in the revised version of NCOMMS-21-47278 (in blue)**

Dear Referees,

Many thanks for taking the time to evaluate our manuscript once again and for the constructive criticism of our study.

Referee # 1

The authors have mostly answered the issues I raised in the earlier review. However, I still think that it might not reach the criteria of this journal, Nat. Commun., because of the low technical novelty.

We thank the Reviewer for assessing the revised version of our manuscript. As indicated in the previous rebuttal, we agree in that other DNA base editors have been described and published recently. We hope that the toolset described in our article will be adopted by researchers in synthetic biology, metabolic engineering, and microbiology—making *Nature Communications* the appropriate platform to share our results and the materials with such a diverse scientific community.

Referee # 2

The authors have substantially improved the quality of their manuscript by incorporating the suggestions made by the reviewers, adding supplementary data that was missing from the initial submission of the manuscript and conducted extra experiments to demonstrate the genome-wide off-targeting effects of their base-editor. One last concern is that I could not find File S5 on the submission portal of the revised manuscript, but perhaps this is just a technical issue (file size too large)?

Thanks for the supportive evaluation of our article. We have made sure that all supplementary files are included in the updated submission of the paper.